# Compositionality in Time Series: A Proof of Concept using Symbolic Dynamics and Compositional Data Augmentation

**Michael Hagmann**                                          *hagmann@cl.uni-heidelberg.de*
*Department of Computational Linguistics*
*Heidelberg University, Germany*

**Michael Staniek**                                          *staniek@cl.uni-heidelberg.de*
*Department of Computational Linguistics*
*Heidelberg University, Germany*

**Stefan Riezler**                                          *riezler@cl.uni-heidelberg.de*
*Department of Computational Linguistics*
*and Interdisciplinary Center for Scientific Computing (IWR)*
*Heidelberg University, Germany*

**Reviewed on OpenReview:** *https://openreview.net/forum?id=msIO2LXVJX*

## Abstract

This work investigates whether time series of natural phenomena can be understood as being generated by sequences of latent states which are ordered in systematic and regular ways. We focus on clinical time series and ask whether clinical measurements can be interpreted as being generated by meaningful physiological states whose succession follows systematic principles. Uncovering the underlying compositional structure will allow us to create synthetic data to alleviate the notorious problem of sparse and low-resource data settings in clinical time series forecasting, and deepen our understanding of clinical data. We start by conceptualizing compositionality for time series as a property of the data generation process, and then study data-driven procedures that can reconstruct the elementary states and composition rules of this process. We evaluate the success of this methods using two empirical tests originating from a domain adaptation perspective. Both tests infer the similarity of the original time series distribution and the synthetic time series distribution from the similarity of expected risk of time series forecasting models trained and tested on original and synthesized data in specific ways. Our experimental results show that the test set performance achieved by training on compositionally synthesized data is comparable to training on original clinical time series data, and that evaluation of models on compositionally synthesized test data shows similar results to evaluating on original test data. In both experiments, performance based on compositionally synthesized data by far surpasses that based on synthetic data that were created by randomization-based data augmentation. An additional downstream evaluation of the prediction task of sequential organ failure assessment (SOFA) scores shows significant performance gains when model training is entirely based on compositionally synthesized data compared to training on original data, with improvements increasing with the size of the synthesized training set.

## 1 Introduction

Compositionality describes the systematic capacity of a system to generate an unbounded number of valid outputs based on novel combinations from a finite set of elementary components. This capacity is considered one of the main pillars of the human ability to generalize to new tasks and situations (Lake et al., 2017). Standard examples for compositionality in human and machine intelligence are natural language and vision,

for example, the capacity to build an infinite number of sentences from a finite vocabulary, or the composition of images from elementary concepts of color, position, and shape. Formal accounts of compositionality arose independently in logic, natural language processing, and computer science, beginning in the 19th century (see Partee (1984), Janssen (2012), or Szabó (2020) for historical overviews). The concept of compositionality currently experiences a renaissance in machine learning under the label of compositional generalization — the ability to systematically generalize to test data that are composed from known components seen in novel combinations. The goal of this recent research strand is to understand and formalize the concept of compositionality, and to inject a compositional inductive bias into state-of-the art generative models, aiming at improved one-shot compositional generalization (Patel et al., 2022) from primitives seen during training to unseen novel combinations at test time.

Our goal is to investigate whether the concept of compositionality can be applied to the less obvious domain of time series, focusing on multivariate and sparse time series of clinical measurements of intensive care patients collected in electronic health records. We start from an algebraic formalization of compositionality as a property of the data generation process which crucially depends on a homomorphism that maps physiological states to clinical observations preserving compositional structure. Compositionality of clinical time series then means that the clinical assessment of a sequence of physiological states is a function of the clinical assessments of its constituents and the way they are sequentially ordered. On the one hand, this interpretation motivates the question of meaningful physiological states and the systematics of their ordering. On the other hand, it motivates a data-driven pipeline that deconstructs the data generation process to empirically detect elementary states and composition rules. An algorithm that discovers elementary components and composition rules will allow us to create synthetic data to alleviate the notorious problem of sparse and low-resource data settings in clinical time series forecasting. Furthermore, we can gain a theoretical understanding of the compositional data generation process underlying time series data from a domain adaptation perspective. The intuition is to infer that original time series data and compositionally synthesized data have the same distribution if domain adaptation from synthesized to original data is successful.

The experimental part of this work is built on an empirical pipeline that uses representation learning, in particular clustering of learned representations of time series subsequences (Ghaderi et al., 2023; Ma et al., 2019), to operationalize a notion of latent classes representing meaningful physiological states, for example, healthy or unhealthy states of certain organ systems. Mapping time series to symbolic representations makes them amenable to compositional methods developed for natural language processing (NLP). Here we apply a simple but entirely data-driven approach to inducing compositional structure from symbolic representations of time series (Andreas, 2020). This algorithm implements the distributional principle (Firth, 1957) to exchange subsequences that occur in the same context to yield other valid sequences.

The theoretical contribution of this work is to motivate empirically testable criteria for the compositionality of the data generation process of time series in domain adaptation theory (Ben-David et al., 2006; 2010b;a; Ben-David & Urner, 2014). We exploit the assumption that successful domain adaptation requires a small distance between source distribution (in our case, the distribution underlying compositionally synthesized data) and target distribution (in our case, original time series data), to infer a small distance between the underlying distributions of synthesized and original data from empirically testing the success of domain adaptation from synthetic to original data. Our first test evaluates compositionally synthesized time series by analyzing their utility for the training of a time series forecasting (TSF) model. Our experiments demonstrate comparable test set performance of models trained on compositionally synthetic data to models trained on the original data for MIMIC-III (Johnson et al., 2016) and eICU (Pollard et al., 2018)) data sets. Our second test compares the use of compositionally synthesized data against original time series data as test data in TSF tasks. Our empirical results show that both test sets yield a similar test set performance for a TSF model trained on original time series data. Furthermore, we find that compositionally synthesized data is much closer to the original data than synthetic data created by a non-compositional data augmentation algorithm (Yun et al., 2019).

In sum, the contributions[1] of our work are as follows: We present an analysis based on symbolic representations and compositional structure that allows an understanding of clinical time series as symbols that

---

[1] Code to reproduce the experiments described in this paper is available at `https://github.com/StatNLP/tmlr_2025_compositional_time_series`

are emitted by an ordered sequence of physiological states. The underlying algorithm allows us to create synthetic training and test data that are on par with the original data in a real-world clinical time series forecasting task and show increasing improvements in the size of synthesized data in downstream tasks. This circumvents the notorious problem of sparse and low-resource data settings in clinical TSF. Furthermore, we present empirically testable criteria for compositionality rooted in domain adaptation theory, allowing broader claims on the compositionality of the data generation process underlying general time series data.

## 2 Related Work

**Symbolic Compositional Structure in NLP.** A connection of our work to NLP can be drawn by viewing natural language sentences as discrete time series of symbols, and by considering the task of next word prediction in language modeling as a TSF task. The central models developed in symbolic NLP — the dominant paradigm in 20th century NLP — were all explicit symbolic generative processes allowing to construct an infinite number of sentences from a finite alphabet (vocabulary) and a finite recursive device (grammar)[2]. Since NLP research has long departed from the symbolic paradigm, the central question in current research on compositional generalization in NLP is consequently an investigation of the compositional skills of non-symbolic neural network architectures. One research strand on compositionality in non-symbolic NLP focuses on the creation of benchmark datasets to evaluate the compositional generalization abilities of neural networks (Lake & Baroni (2018); Keysers et al. (2020); Kim & Linzen (2020), inter alia). Most such datasets are based on the NLP task of semantic parsing, where the components and composition rules are obvious and can be clearly defined. Compositionality is then quantified by measuring accuracy on test sets with a similar component distribution, but different compound distribution. Another research strand focuses on directly injecting a compositional inductive bias into neural sequence models (Russin et al. (2020); Huang et al. (2024); Sartran et al. (2022), inter alia). Our approach directly builds on work on compositional data augmentation Andreas (2020); Akyürek et al. (2021); Qiu et al. (2022) that aims to provide a compositional inductive bias to state-of-the-art sequence learning models by adding compositionally generated data to their training sets.

**Disentangled Representation Learning in Vision.** The goal of disentangled representation learning in vision is to separate informative factors such that a change in a single latent factor leads to a change in a single factor in the learned representation (Bengio et al., 2013). The state-of-the-art models in this area are auto-encoders that directly aim to reconstruct the generative factors of variation (Higgins et al. (2017); Montero et al. (2021); Xu et al. (2022), inter alia). The generalization abilities of these models are usually tested by the task of reconstructing compositionally generated test data that include combinations of generative factors that were not seen during training. However, it has been shown that unsupervised disentangled representation learning is impossible without inductive biases on both models and data (Locatello et al., 2019), and that increased disentanglement does not increase generalization capabilities, especially if one moves away from a simple artificial data set to real-world data (Schott et al., 2022; Montero et al., 2022). Wiedemer et al. (2023) formalize compositionality as a property of the data generation process, assuming the composition function, as well as the latent description of the observations to be known, effectively reducing the learning task to a reconstruction of the component functions. In contrast to this work, we cannot formalize compositional generalization as a direct reconstruction problem since neither latent factors nor the composition function are known for time series. Instead, we indirectly test compositional generalization by evaluating the utility of compositionally synthesized data as training and test data in real-world TSF tasks.

**Domain Adaptation.** Our work takes crucial inspiration from the work of Ben-David et al. (2006; 2010b;a); Ben-David & Urner (2014) to motivate empirically testable criteria for compositionality in domain adaptation theory. Starting from theoretical results that identify necessary and sufficent conditions for a successful DA, we created two experimental setups that allow us to infer a small distance between synthetic and original data based on the distance between expected risks estimates. Similar evaluation setups of training and testing on original and synthetic data have been used by Esteban et al. (2017), however,

---

[2]Symbolic approaches dominated the NLP fields of syntax and semantics, starting with Chomsky (1957) and Montague (1970), respectively, and are still relevant in formal language theory in computer science (starting with Chomsky (1959)).

without a theoretical motivation. Furthermore, our work shows how to generate a synthetic data distribution from an original sample with the properties of matching the original distribution on the level of elementary components, but differing from it at the level of compounds. Our results can be seen as strong hints at the potential of a formal study of compositional data synthesization, with the goal of a better understanding of its generalization properties.

**Data Augmentation.** Data augmentation has grown to an important subfield of machine learning on its own, with a substantial body of work already being devoted to data augmentation for time series (Yoon et al., 2019; Pei et al., 2021; Wen et al., 2021). We would like to stress that the goal of our work is not to present an optimal data augmentation method for time series, but to analyze time series from the perspective of compositional data generation, and to use empirically testable criteria on compositional data synthesization as a proof-of-concept for the validity of this perspective. Insights into the compositional nature of time series can only be gained systematic ablation studies where the contribution of compositionality can be isolated, removed and replaced by random sampling. This is what CutMix (Yun et al., 2019) provides in our case — a randomized variant of our compositional data augmentation method. Mixed-based data augmentation approaches have been successfully applied to various machine learning tasks (Cao et al., 2024), including the area of (physiological) time series data (Guo et al., 2023; Yang & Desell, 2022). Our work provides empirical evidence for an advantage of augmentation techniques that mimic a compositional data generation processes over randomization-based data augmentation.

**Symbolic Dynamics.** Research on transforming real-valued time series into symbolic representations has a long history (see, for example, Williams (2004) for an overview), where the central application is the analysis of dynamical systems (Lind & Marcus, 1995). Since the number of possible symbol assignments grows exponentially with the dimension of the time series, the traditional approach of partitioning the multidimensional phase space spanned by the input variables of a time series into finitely many pieces and then labeling each partition by a specific symbol is only feasible for very low dimensional time series. This problem is overcome by representation learning approaches that first map time series into an embedding space (whose dimensionality can be controlled), where clustering methods are then applied to partition the space (Ma et al., 2019; Ghaderi et al., 2023). We employ the latter techniques to learn a symbolic vocabulary that contains the elementary components of our compositional data synthesization process, and compare it to randomized version of traditional symbolic dynamics.

**Pattern Recognition in Time Series Analysis.** Work on time series motifs aims at the detection of elementary components of time series by identifying short segments that repeat themselves approximately the same within one larger time series (Patel et al., 2002; Schäfer & Leser, 2022). This is orthogonal to our problem of identifying elementary components as time series segments that appear in similar contexts across different time series. Furthermore, detection of motifs in time series is mostly applied to waveform data, e.g., quasi-continuous vital signals such as ECG recordings, whereas our framework is applied to vital measurements where at most one datapoint is taken per hour.

## 3 Compositionality in Time Series

### 3.1 Compositional Data Generation

We conceptualize compositionality as a property of the data generating process, following an algebraic formalization of compositionality (Montague, 1970; Partee, 1984; Szabó, 2020).

**Definition 1.** (Compositional data generation.) Let $(\mathcal{Z}, C_{\mathcal{Z}})$ be an algebraic structure of latent states where $C_{\mathcal{Z}}$ is called the latent state composition function, and let $(\mathcal{X}, C_{\mathcal{X}})$ be an algebraic structure (of the same type) of observations where $C_{\mathcal{X}}$ is called the observation composition function. Furthermore, let $\varphi \colon \mathcal{Z} \to \mathcal{X}$ be a homomorphism that maps latent states to observations. We call a data generating process $f : \mathcal{Z} \to \mathcal{X}$ that satisfies

$$f(z_1, \ldots, z_K) = \varphi(C_{\mathcal{Z}}(z_1, \ldots, z_K)) = C_{\mathcal{X}}(\varphi(z_1), \ldots, \varphi(z_k)) \tag{1}$$

a *compositional data generation process.*

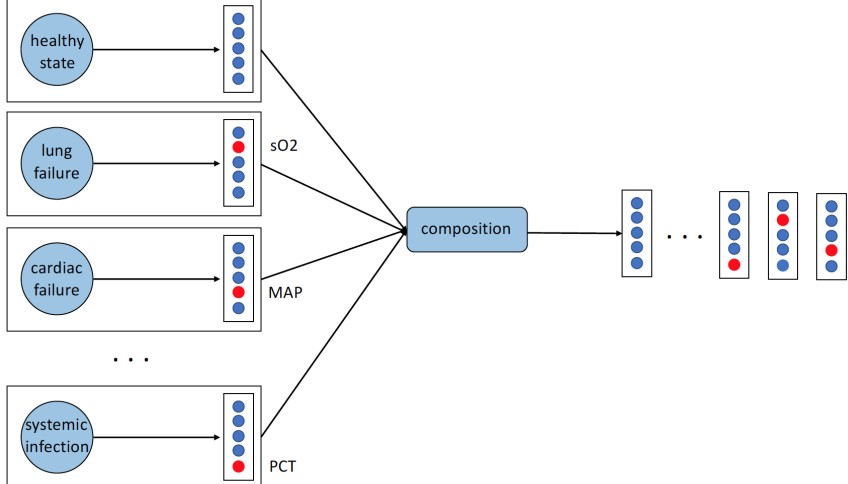

Figure 1: Conceptualization of compositional data generation process for time series of clinical variables: Physiological states are mapped to observations of clinical measurements which are composed into a full time series. Healthy physiological states are realized by observations consisting of default measurements of vital signals or lab measurements. Physiological states representing a failure of organ systems like heart or lung are realized by observation vectors where the measurements of mean arterial pressure (MAP) or oxygen saturation (sO2) are out of a range defined as healthy, or a systemic infection causing an elevated procalcitonin (PCT) measurement. The clinical assessment of this example can be interpreted as a clinical time series of a septic patient with multiple organ system failure, starting with healthy measurements, which are followed by indicators of lung failure and cardiac failure, consequent to a indicators of a systemic infection.

The data generating process of clinical time series can be understood as compositional by the following definition. Let latent states be physiological states of patients, $C_{\mathcal{Z}}$ be the progression of a patient through these states, observations be clinical measurements, and $C_{\mathcal{X}}$ be the temporal regularities within time series. Then Equation 1 can be interpreted as follows:

> The clinical assessment of a sequence of physiological states is a function of the clinical assessments of its constituents and the way they are sequentially ordered.

The right-hand side form of Equation 1 can be illustrated by the generative process shown in Figure 1. This process starts from physiological states of patients, which emit observable components from the space of clinical measurements, which are ordered over time into a full sequence of clinical measurements. The left-hand side of Equation 1 motivates a deconstruction of this process into elementary representations of latent states, which are composed to a sequence of latent states that is mapped to a sequence of observations.

This deconstruction process builds the basis of the experimental work presented in this paper and is illustrated in Figure 2: Starting from input data of multivariate clinical time series, we extract a real-valued vector representation of subsequences from the hidden states of a TSF model, which is then fed into a k-means clustering algorithm (Lloyd, 1982) that allows us to assign a symbolic representation to time series based on the cluster membership of its subsequences. These symbolic representations are the input for an entirely data-driven compositional data augmentation algorithm (Andreas, 2020) that results in an implicit set of rules allowing us to synthesize novel time series in a compositional manner. Each component of this pipeline will be described in more detail below.

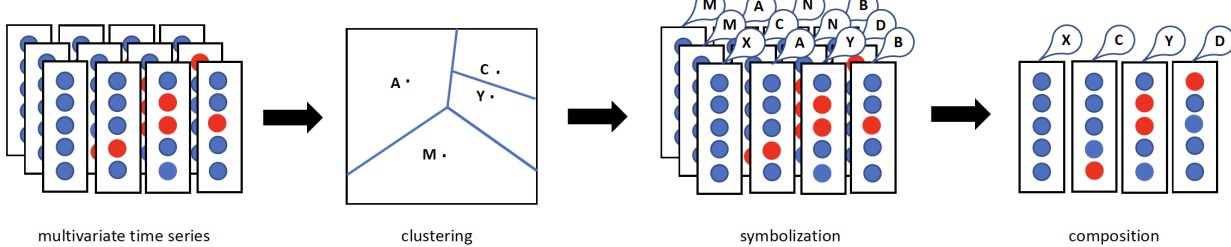

Figure 2: Empirical deconstruction of multivariate time series for compositional data synthesization: First, subsequences of time series are clustered and each cluster is assigned a symbolic label. Based on symbolic representations of time series sub-sequences, compositional data augmentation algorithms can be applied to synthesize full time series in a compositional manner.

### 3.2   A Domain Adaptation Perspective on Compositionality

Viewing the above sketched compositional data augmentation process from a domain adaptation (DA) perspective allows us to arrive at two empirically testable conditions for compositionality. These tests exploit the fact that the training and test distribution of a model are not identical in DA scenarios. For both of our empirical tests, one of these distributions will be synthetically created by a compositional algorithm, and the other distribution will be the original distribution whose compositional status is unknown. The compositionality of the data generation process underlying the original time series data can then be inferred from successful DA from compositionally synthesized to original data.

Our first test is based on theoretical results established by Ben-David et al. (2006; 2010b;a), which provide two jointly necessary and sufficient conditions for successful domain adaptation, one of which is a large enough similarity between training and target distributions. We leverage these results to motivate an empirically testable criterion based on the following rationale. We apply a learning algorithm to both the original training data and compositionally synthesized training data and evaluate the trained models on the original test data. If the estimated expected risks are identical, DA is considered successful and according to the results established by Ben-David et al. (2006; 2010b;a), we can conclude that both distributions must be identical. Thus, we can conclude that the original data generating process must be compositional.

Our second test is also grounded in the theory of DA and leverages a new result that extends an observation by Ben-David & Urner (2014). Our Theorem 1 relates the proximity of the expected risks of a model on two data distributions to the proximity of these distributions. We leverage this theorem to motivate a second empirically testable criterion based on the following rationale. We evaluate a trained learner on both the original test set and a compositionally synthesized test set, and estimate the ratio of the corresponding expected risks. As established by Theorem 1, the magnitude of this ratio allows drawing a conclusion about the proximity between the synthetic and original distributions. If this ratio is one, both distributions are identical, and hence the original data generating process must be compositional.

In the following, we briefly summarize the theoretical concepts of DA theory, especially the necessary and sufficient conditions for successful DA.

Let $\mathcal{X}$ be a domain, $\mathcal{Y}$ be a co-domain, and $\mathcal{P}$ and $\mathcal{Q}$ be distributions over $\mathcal{X} \times \mathcal{Y}$. Furthermore, let $\mathcal{P}_\mathcal{X}$ and $\mathcal{Q}_\mathcal{X}$ denote the respective marginal distributions on $\mathcal{X}$. Let $\mathcal{H} \subseteq \mathcal{Y}^\mathcal{X}$ be a hypothesis class of functions $h$, and let $\ell(y, \hat{y})$ be a loss function on the target labels $y$ and the predicted labels $\hat{y} = h(x)$. We can now define the concepts of a domain adaptation learner (Definition 2), and a concept quantifying domain adaptation learnability (Definition 3).

**Definition 2.** (Domain adaptation learner.) We call a learning algorithm $A$ that receives training samples from $\mathcal{Q}$ but whose expected risk is evaluated on $\mathcal{P}$ a *(conservative) domain adaptation learner*.

**Definition 3.** (($\epsilon, \delta$)-learnability.) Let $\mathcal{P}$ and $\mathcal{Q}$ be distributions on $\mathcal{X} \times \mathcal{Y}$ with common support, $\mathcal{H}$ a hypothesis class, and $A$ be a domain adaptation learner. We say that $A$ ($\epsilon, \delta$)-*learns $\mathcal{P}$ from $\mathcal{Q}$ relative to $\mathcal{H}$*, if for chosen $\epsilon, \delta > 0$ there exists an $n \in \mathbb{N}$ such that when provided a training sample $T$ of size $n$ obtained

from $\mathcal{Q}$, with probability of at least $1 - \delta$ (over the sample space), the expected risk with respect to $\mathcal{P}$ of the obtained classifier $h_T := A(T)$ does not exceed the expected risk $\mathbb{E}_{\mathcal{P}}[\mathcal{H}] := \inf_{h \in \mathcal{H}} \mathbb{E}_{\mathcal{P}}[\ell(y, h(x))]$ of the best classifier on $\mathcal{P}$ by more than $\epsilon$:

$$\Pr_{T \sim \mathcal{Q}^n} \left( \mathbb{E}_{\mathcal{P}}[\ell(y, h_T(x))] - \mathbb{E}_{\mathcal{P}}[\mathcal{H}] \le \epsilon \right) \ge 1 - \delta. \tag{2}$$

Obviously DA works well if $\mathbb{E}_{\mathcal{P}}[\ell(y, h_T(x))] - \mathbb{E}_{\mathcal{P}}[\mathcal{H}]$ can be bounded by a small $\epsilon$, with high probability (small $\delta$) for feasible sample sizes $n$. A further popular assumption in the context of DA is the covariate shift assumption:

**Definition 4.** (Covariate shift.) Let $\mathcal{P}$ and $\mathcal{Q}$ be distributions on $\mathcal{X} \times \mathcal{Y}$ with common support. Then $\mathcal{P}$ and $\mathcal{Q}$ satisfy the covariate shift assumption if the conditional distributions $\mathcal{P}_{\mathcal{Y}|\mathcal{X}}$ and $\mathcal{Q}_{\mathcal{Y}|\mathcal{X}}$ are identical.

The covariate shift assumption is a rather weak requirement that only demands that the stochastic relation between $x$ and $y$ be identical for $\mathcal{Q}$ and $\mathcal{P}$. This assumption is only a necessary, but not a sufficient, condition for a successful DA. This fact is exemplified by the following thought experiment. Let us assume that we trained a model on $\mathcal{Q}$, and that the difference between $\mathcal{Q}$ and $\mathcal{P}$ is such that $\mathcal{Q}_{\mathcal{X}}$ places a lot of mass on inputs that the learner predicts well, and less on inputs where the learner performs poorly. In addition, assume that the situation is exactly the opposite for $\mathcal{P}_{\mathcal{X}}$. Then the expected risk with respect to $\mathcal{Q}$ will be small, but the expected risk with respect to $\mathcal{P}$ will be large. This thought experiment illustrates the need to put a constraint on the difference between $\mathcal{P}_{\mathcal{X}}$ and $\mathcal{Q}_{\mathcal{X}}$. Since the seminal work of Ben-David et al. (2006), it is common to express this difference in terms of the so-called $\mathcal{A}$-distance:

**Definition 5.** ($\mathcal{A}$-distance.) Let $\mathcal{P}_{\mathcal{X}}$ and $\mathcal{Q}_{\mathcal{X}}$ be two distributions on $\mathcal{X}$ and $\mathcal{A} \subseteq 2^{\mathcal{X}}$ such that each set in $\mathcal{A}$ is measurable with respect to both distributions. Then the $\mathcal{A}$-distance between $\mathcal{P}_{\mathcal{X}}$ and $\mathcal{Q}_{\mathcal{X}}$ is

$$\mathrm{d}_{\mathcal{A}}(\mathcal{P}_{\mathcal{X}}, \mathcal{Q}_{\mathcal{X}}) := 2 \sup_{A' \in \mathcal{A}} |\mathcal{P}_{\mathcal{X}}(A') - \mathcal{Q}_{\mathcal{X}}(A')| \tag{3}$$

It obviously is sufficient to consider only those domain subsets where the potential hypotheses predict different outputs $\mathcal{H}\Delta\mathcal{H} := \{\{x \in \mathcal{X} \mid h(x) \ne h'(x)\} \mid h, h' \in \mathcal{H}\}$. Since we select a hypothesis that should perform well with respect to $\mathcal{P}$ based solely on the information from $\mathcal{Q}$, we need to assume that there exists a hypothesis $h^*$ that performs well on both distributions. This hypothesis has been named the low-error joint predictor by Ben-David et al. (2006):

**Definition 6.** (Low-error joint predictor.) We call a hypothesis $h^*$ a low-error joint predictor, if

$$\mathbb{E}_{\mathcal{P}}[\ell(y, h^*(x))] + \mathbb{E}_{\mathcal{Q}}[\ell(y, h^*(x))] \approx \mathbb{E}_{\mathcal{P}}[\mathcal{H}] + \mathbb{E}_{\mathcal{Q}}[\mathcal{H}] \tag{4}$$

were $\mathbb{E}_{\mathcal{P}}[\mathcal{H}] := \inf_{h \in \mathcal{H}} \mathbb{E}_{\mathcal{P}}[\ell(y, h(x))]$ and $\mathbb{E}_{\mathcal{Q}}[\mathcal{H}] := \inf_{h \in \mathcal{H}} \mathbb{E}_{\mathcal{Q}}[\ell(y, h(x))]$.

It has been shown by Ben-David et al. (2006; 2010b) that the combination of a small distance $\mathrm{d}_{\mathcal{H}\Delta\mathcal{H}}(\mathcal{P}_{\mathcal{X}}, \mathcal{Q}_{\mathcal{X}})$ and the existence of $h^*$ is necessary and sufficient for $A$ to be a $(\epsilon, \delta)$-learner (making the covariate shift assumption redundant if both of the former conditions hold).

For our first proposed test, we train a model once on the original training data to get an estimate of $\mathbb{E}_{\mathcal{P}}[\mathcal{H}]$, and also on compositionally synthesized training data to get an estimate for $\mathbb{E}_{\mathcal{P}}[\ell(y, h_T(x))]$. Calculating the difference between these estimates allows us to estimate an approximate magnitude for $\epsilon$. If this magnitude is small, DA was successful. This allows us to infer a small $\mathcal{A}$-distance between the compositionally synthesized and the original data distributions. Hence, we conclude that the original data generating process must be compositional as well.

If $h^*$ does not exist, we can still test the similarity of $P$ and $Q$ under the covariate shift assumption based on the expected risk ratio. The following theorem takes inspiration from Ben-David & Urner (2014) who observed that a bounded ratio between $\mathcal{P}_{\mathcal{X}}$ and $\mathcal{Q}_{\mathcal{X}}$ implies a bounded ratio of the expected risks calculated with respect to $\mathcal{P}$ and $\mathcal{Q}$ for any learner. A proof is given in Appendix A.3.

**Theorem 1.** (Bounded expected risk and density.) Let $\mathcal{P}$ and $\mathcal{Q}$ be two absolutely continuous probability distributions on $\mathcal{X} \times \mathcal{Y}$ for which the covariate shift assumption holds such that $f_{\mathcal{P}}(x, y) = f(y \mid x) f_{\mathcal{P}}(x)$ and

$f_{\mathcal{Q}}(x, y) = f(y \mid x)f_{\mathcal{Q}}(x)$ are the corresponding densities. Further let $\mathcal{H}$ be a hypotheses class of learners, $\ell \geq 0$ be a loss function, and $\underline{C}, \overline{C} \in \mathbb{R}^+$. Then,

$$\forall x \in \mathcal{X} \colon \underline{C}f_{\mathcal{Q}}(x) \leq f_{\mathcal{P}}(x) \leq \overline{C}f_{\mathcal{Q}}(x) \iff \forall h \in \mathcal{H} \colon \underline{C}\mathbb{E}_{\mathcal{Q}}\left[\ell\left(x, h(x)\right)\right] \leq \mathbb{E}_{\mathcal{P}}\left[\ell\left(x, h(x)\right)\right] \leq \overline{C}\mathbb{E}_{\mathcal{Q}}\left[\ell\left(x, h(x)\right)\right]. \tag{5}$$

Further $\underline{C} \leq 1$ and $\overline{C} \geq 1$.

Under the rather unproblematic assumption that $\mathbb{E}_{\mathcal{Q}}\left[\ell\left(x, h(x)\right)\right] > 0$, and by defining $\frac{f_{\mathcal{P}}(x)}{f_{\mathcal{Q}}(x)} = 1$ for points in $\mathcal{X}$ whose densities are zero, we obtain bounds on the ratios of densities and expected risks:

$$\forall x \in \mathcal{X} \colon \underline{C} \leq \frac{f_{\mathcal{P}}(x)}{f_{\mathcal{Q}}(x)} \leq \overline{C} \iff \forall h \in \mathcal{H} \colon \underline{C} \leq \frac{\mathbb{E}_{\mathcal{P}}\left[\ell\left(x, h(x)\right)\right]}{\mathbb{E}_{\mathcal{Q}}\left[\ell\left(x, h(x)\right)\right]} \leq \overline{C}. \tag{6}$$

We utilize this relation to define a second empirically testable criterion. We evaluate a trained learner on the original test set to estimate $\mathbb{E}_{\mathcal{P}}\left[\ell\left(x, h(x)\right)\right]$, and on the compositionally synthesized test set to estimate $\mathbb{E}_{\mathcal{Q}}\left[\ell\left(x, h(x)\right)\right]$. The ratio of these estimates is an estimate of the corresponding expected risk ratio. This allows us to assess whether $\underline{C}$ and $\overline{C}$ are close to one, and by the above result, to infer whether $f_{\mathcal{P}}(x, y)$ and $f_{\mathcal{Q}}(x, y)$ are close. If so, we conclude that the original data generating process is compositional as well.

## 4 Methods

### 4.1 Symbolic Dynamics

The empirical deconstruction process of multivariate time series shown in Figure 2 first needs to identify elementary components that can in a second step be used to synthesize new time series in a compositional manner. Since the composition technique used in the second step relies on discrete representations of sub-segments of time series, we need to transform sub-sequences of real-valued vectors into symbolic representations. The central concept in this context is the notion of a symbol space given in Definition 7.

**Definition 7.** (Symbol space.) Let $(\mathcal{X}, \mathrm{d})$ be a finite dimensional metric space with metric d and $c_1, \ldots, c_k \in \mathcal{X}$. Then the partition of $\mathcal{X}$ given by:

$$S_i = \left\{ x \in \mathcal{X} \mid c_i = \underset{j=1,\ldots,k}{\arg\min}\left(\mathrm{d}(x, c_j)\right) \right\} \tag{7}$$

for all $i = 1, \ldots, k$ is called the symbol space of $\mathcal{X}$ with symbols $S_i$ and centroids $c_i$.

In order to transform a time series into a chain of symbols, we break an $n$-dimensional multivariate time series of length $T$ into consecutive non-overlapping blocks of length $\Delta$ subject to $T \mod \Delta = 0$. Next we map these blocks to their corresponding symbol in the symbol space and arrange them in the same order as the blocks. The domain of this mapping can be either the original input space $\mathcal{M}_{n,\Delta}$, or the space of the learned neural block representations. In the first case, a straightforward application of traditional symbolic dynamics methods (Lind & Marcus, 1995; Williams, 2004) would require an alphabet size that grows exponentially with the number of input dimensions. This is infeasible even for multivariate time series of around 100 clinical variables. An approach to circumvent this problem is to randomly select a feasible number of centroids in the input domain $\mathcal{M}_{n,\Delta}$.

In the second case, we exploit the representations learned by a neural network and apply $k$-means[3] clustering to create a symbolic representation of the time series with a computational learning cost that is linear in the embedding dimension. The learned representations consist of the hidden states of the encoder of the TSF Transformer described in Section 5.1. This Transformer model was trained to predict three future hours based on the current three hours on the training set, with hyperparameter settings as described in Appendix A.1 (except the hidden size set to 50).

|  | M | *A* | N | *B* |
|---|---|---|---|---|
| training data | *She* | *picks* | *the suitcase* | *up* |
|  | M | *C* | N | *D* |
|  | *She* | *puts* | *the suitcase* | *down* |
|  | X | *A* | Y | *B* |
|  | *He* | *picks* | *the box* | *up* |
| augmentation | X | *C* | Y | *D* |
|  | *He* | *puts* | *the box* | *down* |

- Identify interchangeable *fragments* as (discontinuous) symbol sequences that appear in similar contexts (*environments*) in the training data, for example, $A \ldots B$ and $C \ldots D$ in context $M \ldots N$.

- Find additional contexts in which interchangeable fragments appear, forming a *template* for substituting in another fragment, for example, context $X \ldots Y$ for fragment $A \ldots B$.

- Augment the data with a synthesized training example by exchanging fragments in a template, for example, substitute $C \ldots D$ for $A \ldots B$ in context $X \ldots Y$.

Figure 3: Compositional data synthesization procedure for symbol sequences adapted from Andreas (2020). Symbol sequences $A \ldots B$, $C \ldots D$, $M \ldots N$, and $X \ldots Y$ are generic and can be discontinuous. An illustration with natural language sentences if given in the respective second lines.

## 4.2 Data-Driven Compositional Data Synthesization

Andreas (2020) presented a linguistically motivated data-driven approach to induce compositional structure in symbol sequences in NLP. This algorithm can be readily applied to induce compositional structure from symbolic representations of time series. The key inspiration of the algorithm is the *distributional structure* of language (Harris, 1954) according to which "You shall know a word by the company it keeps!" (Firth, 1957). This principle is grounded in the fact that words that are used and occur in the same contexts tend to purport similar meanings. The method of Andreas (2020) exploits the distributional principle to generate novel valid sequences by exchanging subsequences with similar meanings into new contexts.[4] An illustrative example of the distributional structure of language and how the method uses it is given in Figure 3.

In the following, we will present a formal account of the compositional data synthesization method illustrated in Figure 3. We will introduce the notions of fragment, template, and environment of a sequence, and give a formal definition of the synthesization process and provide pseudo-code for the algorithm used in our work (see Algorithm 1 in Appendix A.4).

**Definition 8.** (Sequence.) A sequence of length $k$ is a $k$-tuple whose elements are called symbols.

**Definition 9.** (Fragment.) Given a sequence $s$ and a set of indices $I_{\text{frg}} \in 2^{\{1,\ldots,k\}}$. A fragment is the tuple of subsequences of $s$ given by consecutive indices in $I_{\text{frg}}$.

**Definition 10.** (Template.) Given a sequence $s$ with length $k$ and a pair of index sets $I_{\text{frg}} \in 2^{\{1,\ldots,k\}}$ and $I_{\text{tpl}} := \{1, \ldots, k\} \setminus I_{\text{frg}}$, a template is defined as the tuple whose symbols are identical to those of $s$ for all the indices of $I_{\text{tpl}}$, and a slot symbol ★ for all indices in $I_{\text{frg}}$ where consecutive slot symbols are reduced to one.

**Definition 11.** (Environment.) Given a template $t = (t_1, \ldots, t_k)$ and a window size $w$. Let $I_{\text{env}} := \{i \in \{1, \ldots, k\} \mid ★ \in (t_{\max(1,i-w)}, \ldots, t_{\min(k,i+w)})\}$. The corresponding subsequence of $t$ given by $I_{env}$ is called the environment of $t$ (with respect to the window size $w$).

---

[3]We used the scikit-learn (Pedregosa et al., 2011) version of $k$-means clustering, implementing Lloyd's standard algorithm (Lloyd, 1982) by default.

[4]Note that this algorithm works on historical data and is but one possible approximation of the original data generation process assumed to underlie (clinical) time series. We consider its purely data-driven nature and its impartiality towards the underlying compositional process an advantage for the current exposition.

**Definition 12.** (Insert). Given a template $t = (t_1, \ldots, t_k)$ and a fragment $f = (f_1, \ldots, f_m)$ where $\mathrm{card}(f) = \mathrm{card}\,(t_i \in t \mid t_i = \bigstar)$, and let $i_1, \ldots, i_m$ be the slot indices of $t$, then:

$$\mathrm{insert}\colon (t_1, \ldots, t_k) \times (f_1, \ldots, f_m) \to (s_1, \ldots, s_n)$$

$$(t_1, \ldots, t_k) \times (f_1, \ldots, f_m) \mapsto \mathrm{flatten}\left(\left(s_i = \begin{cases} f_j, & i = i_j \\ t_i, & t_i \neq \bigstar \end{cases}\right)_{i=1}^{k}\right)$$

combines $t$ and $f$ by replacing the slot symbols of $t$ with the symbols of the corresponding subsequences of $f$ to a sequence $s$.

**Definition 13.** (Compositional data synthesization.) Let there be a fragment $f$ and two non-identical sequences $s_a$ and $s_c$ such that $s_a = \mathrm{insert}(t_a, f)$ and $s_c = \mathrm{insert}(t_c, f)$. Furthermore, let there be a sequence $s_b = \mathrm{insert}(t_b, f_b)$ such that $f_b \neq f$ and $\mathrm{environment}(t_a, w) = \mathrm{environment}(t_b, w)$. Then $s_{\mathrm{syn}} := \mathrm{insert}(t_c, f_b)$ is a compositionally created synthetic symbol sequence that is different to $s_a$, $s_b$ and $s_c$.

In order to translate a sequence $s = \mathrm{insert}(t, f)$ into a time series, we choose a time series whose symbolization is identical to the sequence $s$. Because symbolization is index-preserving, we can directly map time series sections to $t$ and $f$ by simply replacing their symbols with the time series blocks of the same index. To compose a synthetic time series $x_{\mathrm{syn}} := \mathrm{insert}(t_c, f_b)$ we obtain time series $x_b$ and $x_c$ with the corresponding symbolizations $s_b$ and $s_c$ and desymbolize $t_c$ and $f_b$ accordingly.

## 5 Experiments

### 5.1 Data, Preprocessing, and Models

In our experiments, we focus on two clinical time series datasets: MIMIC-III (Johnson et al., 2016) and eICU (Pollard et al., 2018). Both databases contain anonymized information from ICU patients, including physiological measurements (e.g., heart rate, blood pressure), medications (e.g., dobutamine, epinephrine), interventions (e.g., intubation), lab test results (e.g., blood cultures,) and clinical notes. We extracted features that were frequently tracked during a patient's stay, namely 131 features for MIMIC-III, and 98 for eICU (a complete list can be found in Appendix A.2). For each patient stay we extracted the first 48 hours, but only took stays into account that had at least one feature measured per hour. Furthermore, we removed patients that did not stay long enough in the ICU. The remaining clinical variables were $z$-score standardized, and the resulting datasets were partitioned into training, validation and test data (see Table 5 in Appendix A.2 for exact numbers). Notably, our data can be characterized as sparse and irregularly sampled (Tipirneni & Reddy, 2022; Horn et al., 2020). This is because some features like heart rate or blood pressure are recorded every 5 minutes, other features like those only available through blood sampling, are available only every 24 hours. We store these sparse multivariate time series in a database of $n$ quadruplets $\mathrm{S} = \{(f_i, t_i, v_i, n_i)\}_{i=1}^{n}$, where $f_i \in F$ is a clinical variable identifier, $t_i \in \mathbb{R}_{\geq 0}$ is a time index, $v_i \in \mathbb{R}$ the observed value of $f_i$ at $t_i$, and $n_i$ the unique stay identifier. Following Staniek et al. (2024) who showed an advantage for compressing long input time series into 24 hourly bins that record the most important observations, we encode each quadruplet $\mathrm{S}$ into a dense representation $x$ where every timestep is a vector of feature values representing one hour. We construct this vector by choosing the first observed value during the represented hour for each feature. If no value was observed, we impute zero which corresponds to the mean value due to standardization of the data. Additionally, a mask indicating whether a value was imputed is generated and appended to the vector. Based on this representation, we define the sparsity of a feature as the relative frequency of imputations. For TSF, we use a Transformer model with an autoregressive decoder that generates an output vector $\hat{y}_t \in \mathbb{R}^{|F|}$ (where $|F|$ is the number of features used). The predicted output $\hat{y}_t$ is a function of the history $\hat{y}_{<t}$ of predicted timesteps until time $t$, the encoded input $x$, and the model parameters $\theta$: $\hat{y}_t = f_\theta(\hat{y}_{<t}, x)$. To perform long-term TSF using the autoregressive setup, the outputs $\hat{y}_t$ from each time step $t = 1, \ldots, T$ are concatenated. We employ a standard Transformer architecture (Vaswani et al., 2017) as our model of choice, the hyperparameters of which can be found in Appendix A.1. The encoder takes as input the first 24 hours, the decoder then generates the next 24 hours. The complete model is trained with masked mean squared error (MSE) (see Appendix A.5).

Table 1: Estimated expected risk difference $\hat{\epsilon}$ between model trained on synthesized data and model trained on original data, evaluated on original test data. Synthetic data are generated non-compositionally (CutMix) or compositionally (CDS). Symbolization is obtained by clustering in input or neural embedding space. Results are averages of models trained from three different random seeds, standard error (SE) in brackets.

| Synthetic distribution | Symbolization | MIMIC-III | eICU |
|:---:|:---:|:---:|:---:|
| | | $\hat{\epsilon}$ (SE) | $\hat{\epsilon}$ (SE) |
| CutMix | | 0.753 (0.015) | 0.911 (0.012) |
| CDS | input | 0.070 (0.015) | 0.064 (0.012) |
| CDS | embd | 0.110 (0.019) | 0.055 (0.015) |

Table 2: Estimated ratio of expected risk on synthesized and original test data for model $h^*$ with best result (MSE$^*$) obtained by training on the original training data.

| Synthetic distribution | Symbolization | MIMIC-III | | eICU | |
|:---:|:---:|:---:|:---:|:---:|:---:|
| | | MSE$^*$ | Ratio | MSE$^*$ | Ratio |
| CutMix | | 10.201 | 1.375 | 7.648 | 1.445 |
| CDS | input | 7.398 | 0.997 | 5.279 | 0.998 |
| CDS | embd | 7.230 | 0.974 | 5.086 | 0.961 |

To generate a synthetic dataset, we first need to assign symbols to the training data, as described in Section 4.1, by either choosing random centroids in the input space (input), or by applying $k$-means clustering to the learned representations (embd), with a variable number of centroids (#Syms). In our experiments, the number of symbols is varied between 40, 80, an 160, with best results on the validation set obtained for the largest symbol size. Furthermore, each symbol represents a time series segment of 3 hours, which we found to be a good compromise between very sparse 1-hour windows where over 90% of the measurements need to be imputed, and coarse grained segments with a better descriptive power of temporal patterns. Last, we apply the compositional data synthesization algorithm described in Section 4.2 to generate compositional synthetic data versions, or use the CutMix algorithm (Yun et al., 2019) directly on the time series to produce non-compositional synthetic data.

## 5.2 Experimental Results

### 5.2.1 Test 1: Utility of Synthesized Data for TSF Training

The first empirically testable criterion described in Section 3.2 infers the compositionality of the original data generating process from the estimated expected risk difference $\hat{\epsilon} \approx \mathbb{E}_{\text{orig}}\left[\ell(y, h_T(x))\right] - \mathbb{E}_{\text{orig}}\left[\mathcal{H}\right]$, where $\mathbb{E}_{\text{orig}}\left[\ell(y, h_T(x))\right]$ is estimated by training a TSF model on synthetic data and evaluate it on the original test data, and $\mathbb{E}_{\text{orig}}\left[\mathcal{H}\right]$ is estimated by training a TSF model on the original time series data and evaluate it on the original test data. The synthetic data can be generated either non-compositional (CutMix) or compositional (CDS) with symbolization obtained by clustering in input or neural embedding space. The estimate of $\hat{\epsilon}$ and its standard error is obtained by training a linear mixed effects model (Demidenko, 2013; Bates et al., 2015; Riezler & Hagmann, 2024) on the evaluation scores of models obtained from three different random seeds for optimization, and another three random seeds for symbolization. The main result is given in Table 1: It shows values of $\hat{\epsilon}$ that are close to zero for CDS training, and an order of magnitude larger for training on randomization-based synthetic data, on both datasets of clinical time series.

### 5.2.2 Test 2: Utility of Synthesized Data as TSF Test Data

Table 2 presents the experimental results for the second empirically testable criterion established in Section 3.2. This test is based on a model $h^*$ that is obtained by training on original time series data and its evaluation on original and synthetic test data. The goal is to infer a uniform bound for the ratio $\mathbb{E}_{\text{syn}}\left[\ell(x, h^*(x))\right]/\mathbb{E}_{\text{orig}}\left[\ell(x, h^*(x))\right]$ of the synthetic and original data distribution based on estimates of the

Table 3: Evaluation of downstream task of prediction of SOFA score on original test set for models trained on different sizes of synthesized data. Results are reported for best performing models (MSE$^*$) with 95% confidence intervals for the estimation of the respective evaluation score on the test set in subscripts.

| Training data | Dataset size | MIMIC-III | eICU |
|---|---|---|---|
| original | 1x | $4.168_{[3.962,4.375]}$ | $2.518_{[2.388,2.647]}$ |
| CDS-input | 1x | $4.066_{[3.854,4.278]}$ | $2.536_{[2.389,2.682]}$ |
| | 5x | $3.560_{[3.358,3.763]}$ | $2.397_{[2.254,2.539]}$ |
| | 10x | $3.332_{[3.148,3.515]}$ | $2.347_{[2.217,2.476]}$ |
| | 20x | $3.194_{[3.020,3.369]}$ | $2.322_{[2.189,2.454]}$ |
| CutMix | 20x | $4.063_{[3.836,4.291]}$ | $2.583_{[2.443,2.723]}$ |

corresponding expected risks $\mathbb{E}_{\mathrm{syn}}\left[\ell\left(x, h^*(x)\right)\right]$ and $\mathbb{E}_{\mathrm{orig}}\left[\ell\left(x, h^*(x)\right)\right]$. We see that the obtained ratios are close to 1 for evaluations on original data and compositionally synthesized data, but not for test data synthesized via the randomization-based CutMix method. This again supports our reasoning of compositionality of the data generation process underlying the original time series data.

### 5.2.3 Evaluation on Downstream Task of SOFA Score Prediction

We performed an experiment where we evaluated training solely on synthesized data on the downstream task of prediction of the sequential organ failure assessment (SOFA) score (Vincent et al., 1996). The SOFA score is defined as the sum of subscores for six organ systems, each ranging from 0-4 and depending for their part on thresholded fundamental clinical variables observed during a 24h window (see Appendix A.6). We added a regression head to our dense encoder, and trained this model on automatically assigned SOFA scores (ranging from 0 to 24). The evaluation was done by computing MSE of the predicted scores. Table 3 shows the results of this downstream evaluation. Assessing statistical significance by non-overlapping confidence intervals, we can attribute significant improvements for SOFA score prediction on MIMIC-III for models trained on compositionally synthesized data (CDS-input) that are 5, 10, or 20 times larger than the original training set. Similar, but smaller nominal improvements, are obtained by compositional data augmentation on the larger eICU dataset. However, on neither dataset, prediction for models trained on randomized-based data synthesization (CutMix) significantly improves over models trained on original data, despite the size of the synthetic dataset being 20 times larger.

### 5.2.4 Further Experimental Evaluation

In Appendix A.7, we present an analysis of the distributional properties of compositionally created data. We find that the Hellinger distance of distributions of original and compositionally synthesized data show the desired properties for compositional data (Keysers et al., 2020): Closeness in unigram space, indicating similar distributions of symbols, and larger distance in higher n-gram space, indicated by dissimilar distributions of compounds. Furthermore, in Appendix A.8, we present a qualitative assessment of the symbolic representations of clinical time series segments, showing that meaningful physiological states can be learned by symbolic representation learning. Lastly, in Appendix A.9, we present a quantitative evaluation with respect to the discriminative score, and a qualitative evaluation according to a PCA visualization, inspired by works on GAN-based data synthesization (Yoon et al., 2019; Pei et al., 2021).

## 6 Conclusion

We presented an investigation of the question whether non-linguistic time series — here time series of clinical measurements — also exhibit the intriguing property of compositionality of natural language sequences, namely the characteristics of being generated by a process that combines elementary components following a compositional rule system. If so, we could analyze clinical time series as a series of subsequences that are emitted by an ordered sequence of physiological states, and create synthetic data for notoriously sparse and low-resource data situations in clinical TSF. We showed that a method from NLP that is based entirely on

distributional properties of data allows us to synthesize novel combinations of elementary time series subsequences with most interesting properties: Training and testing of TSF models on compositionally synthesized time series yields a similar utility than training and testing on original time series data, allowing us to conclude that the data generation process underlying the original time series data can in fact be characterized as compositional. Our experiments are based on applying a pipeline of data-driven symbolic representation learning and compositional data augmentation to clinical time series data, and we find consistent improvements over randomization-based data synthesization for TSF on two clinical time series datasets, and for the downstream task of SOFA score prediction. We show that our empirical tests can be motivated in domain adaptation theory, drawing on a possible inference about the distributions of the data generation process of source and target data from the performance of training and testing models on these data. This theoretical motivation allows us to make broader claims on the compositionality of time series data beyond clinical time series, opening the doors to further research on compositional data augmentation and domain adaptation in general time series modeling tasks.

A limitation of our work is the lack of an embedding into a specific clinical problem, based on problem-specific clinical measurements, and an evidence-based clinical interpretation. This will be an important task for future work.

### Acknowledgments

This research was partially funded by Germany's Excellence Strategy EXC 2181/1 – 390900948 (STRUCTURES). We would like to thank Daniel Durstewitz and Filip Sadlo for fruitful discussions on early ideas leading to this work. Finally, we would like to thank the anonymous reviewers for pointers to related work.

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

# A  Appendix

## A.1  Hyperparameters

Table 4: Hyperparameter settings for training. Best settings chosen on development data are shown in bold face.

| Parameter | MIMIC-III | eICU |
|---|---|---|
| Embedding Size | 128, 256, **512** | **256**, 512, 1024 |
| Hidden Size Encoder | 128, 256, **512** | **256**, 512, 1024 |
| Hidden Size DMS Decoder | 128, 256, **512** | **256**, 512, 1024 |
| Hidden Size IMS Decoder | Output Dimensionality | Output Dimensionality |
| # Encoder Layers | 1, **2** | 1, **2**, 3 |
| # Decoder Layers | **1**, 2 | **1** |
| Learning Rate | **0.0005** | **0.0005** |
| Finetune Learning Rate | **0.0001** | **0.0001** |
| Batch Size | 32 | 32 |
| Attention Heads Encoder | 2, 4, **8** | **8** |
| Attention Heads Decoder | **1**, 2, 4 | **1**,2,4 |
| Dropout | **0.05**, 0.1, 0.2 | **0.05** |
| Epochs | 100 | 600 |
| Patience | 6 | 6 |
| Random Seed | Unixtime variation | Unixtime variation |

## A.2  Data Statistics and Feature Lists

Table 5: Number of patient stays in clinical time series.

| Data split | MIMIC-III | eICU |
|---|---|---|
| train | 12,304 | 49,730 |
| dev | 3,169 | 12,433 |
| test | 3,803 | 3,008 |

Table 6: Feature list for MIMIC-III: Besides the following 131 dynamic variables, only age and gender were extracted. The 15 variables marked with an asterisk are directly used for calculating the SOFA score.

| | | | |
|---|---|---|---|
| ALP | Epinephrine* | LDH | Packed RBC |
| ALT | Famotidine | Lactate | Pantoprazole |
| AST | Fentanyl | Lactated Ringers | Phosphate |
| Albumin | FiO2* | Levofloxacin | Piggyback |
| Albumin 25% | Fiber | Lorazepam | Piperacillin |
| Albumin 5% | Free Water | Lymphocytes | Platelet Count* |
| Amiodarone | Fresh Frozen Plasma | Lymphocytes (Absolute) | Potassium |
| Anion Gap | Furosemide | MBP | Pre-admission Intake |
| BUN | GCS_eye* | MCH | Pre-admission Output |
| Base Excess | GCS_motor* | MCHC | Propofol |
| Basophils | GCS_verbal* | MCV | RBC |
| Bicarbonate | GT Flush | Magnesium | RDW |
| Bilirubin (Direct) | Gastric | Magnesium Sulfate (Bolus) | RR |
| Bilirubin (Indirect) | Gastric Meds | Magnesium Sulphate | Residual |
| Bilirubin (Total)* | Glucose (Blood) | Mechanically ventilated | SBP* |
| CRR | Glucose (Serum) | Metoprolol | SG Urine |
| Calcium Free | Glucose (Whole Blood) | Midazolam | Sodium |
| Calcium Gluconate | HR | Milrinone | Solution |
| Calcium Total | Half Normal Saline | Monocytes | Sterile Water |
| Cefazolin | Hct | Morphine Sulfate | Stool |
| Chest Tube | Heparin | Neosynephrine | TPN |
| Chloride | Hgb | Neutrophils | Temperature |
| Colloid | Hydralazine | Nitroglycerine | Total CO2 |
| Creatinine Blood* | Hydromorphone | Nitroprusside | Ultrafiltrate |
| Creatinine Urine | INR | Norepinephrine* | Urine* |
| D5W | Insulin Humalog | Normal Saline | Vancomycin |
| DBP* | Insulin NPH | O2 Saturation | Vasopressin |
| Dextrose Other | Insulin Regular | OR/PACU Crystalloid | WBC |
| Dobutamine* | Insulin largine | PCO2 | Weight |
| Dopamine* | Intubated | PO intake | pH Blood |
| EBL | Jackson-Pratt | PO2* | pH Urine |
| Emesis | KCl | PT | |
| Eoisinophils | KCl (Bolus) | PTT | |

Table 7: Feature list for eICU: Besides the following 98 dynamic variables, there are 17 static variables covering age, gender, admission information, and ICU type. The 15 variables marked with an asterisk are directly used for calculating the SOFA score. On the right column, there are 35 drug-related variables. Some of them seem redundant due to different hospitals but can not be merged because of different or not standardized concentrations.

| | | |
|---|---|---|
| ALP | Lactate | Amiodarone |
| ALT | Lymphocytes | Dobutamine dose |
| AST | MBP | Dobutamine ratio* |
| Albumin | MCH | Dopamine dose |
| Anion Gap | MCHC | Dopamine ratio* |
| BUN | MCV | Epinephrine dose |
| Base Deficit | MPV | Epinephrine ratio* |
| Base Excess | Magnesium | Fentanyl 1 |
| Basophils | Monocytes | Fentanyl 2 |
| Bedside Glucose | Neutrophils | Fentanyl 3 |
| Bicarbonate | O2 L/% | Furosemide |
| Bilirubin (Direct) | O2 Saturation | Heparin 1 |
| Bilirubin (Total)* | PT | Heparin 2 |
| Bodyweight (kg) | PTT | Heparin 3 |
| CO2 (Total) | PaCO2 | Heparin vol |
| Calcium | PaO2* | Insulin 1 |
| Chloride | Phosphate | Insulin 2 |
| Creatinine (Blood)* | Platelets* | Insulin 3 |
| Creatinine (Urine) | Potassium | Midazolam 1 |
| DBP* | Protein (Total) | Midazolam 2 |
| Eoisinophils | RBC | Milrinone 1 |
| EtCO2 | RDW | Milrinone 2 |
| FiO2* | RR | Nitroglycerin 1 |
| Fibrinogen | SBP* | Nitroglycerin 2 |
| GCS eye* | Sodium | Nitroprusside |
| GCS motor* | Stool | Norepinephrine 1 |
| GCS verbal* | Temperature | Norepinephrine 2 |
| Glucose | Troponin - I | Norepinephrine ratio* |
| HR | Urine* | Pantoprazole |
| Hct | WBC | Propofol 1 |
| Hgb | pH | Propofol 2 |
| INR | | Propofol 3 |
| | | Vasopressin 1 |
| | | Vasopressin 2 |
| | | Vasopressin 3 |

### A.3 Proof for Theorem 1

**Theorem 1.** (Bounded expected risk and density.) Let $\mathcal{P}$ and $\mathcal{Q}$ be two absolutely continuous probability distributions on $\mathcal{X} \times \mathcal{Y}$ for which the covariate shift assumption holds such that $f_\mathcal{P}(x,y) = f(y \mid x)f_\mathcal{P}(x)$ and $f_\mathcal{Q}(x,y) = f(y \mid x)f_\mathcal{Q}(x)$ are the corresponding densities. Further let $\mathcal{H}$ be a hypotheses class of learners, $\ell \geq 0$ be a loss function, and $\underline{C}, \overline{C} \in \mathbb{R}^+$. Then,

$$\forall x \in \mathcal{X}: \underline{C}f_\mathcal{Q}(x) \leq f_\mathcal{P}(x) \leq \overline{C}f_\mathcal{Q}(x) \iff \forall h \in \mathcal{H}: \underline{C}\mathbb{E}_\mathcal{Q}\left[\ell\left(x, h(x)\right)\right] \leq \mathbb{E}_\mathcal{P}\left[\ell\left(x, h(x)\right)\right] \leq \overline{C}\mathbb{E}_\mathcal{Q}\left[\ell\left(x, h(x)\right)\right].$$

Further $\underline{C} \leq 1$ and $\overline{C} \geq 1$.

*Proof.* To establish sufficiency, let us assume that $\forall x \in \mathcal{X}: \underline{C}f_\mathcal{Q}(x) \leq f_\mathcal{P}(x) \leq \overline{C}f_\mathcal{Q}(x)$. Then

$$\forall (x,y) \in \mathcal{X} \times \mathcal{Y}: \underline{C}\underbrace{f(y \mid x)f_\mathcal{Q}(x)}_{=f_\mathcal{Q}(x,y)} \leq \underbrace{f(y \mid x)f_\mathcal{P}(x)}_{=f_\mathcal{P}(x,y)} \leq \overline{C}\underbrace{f(y \mid x)f_\mathcal{Q}(x)}_{=f_\mathcal{Q}(x,y)}$$

Because $\ell$ is non-negative for all $h$, we can conclude by the monotonicity and linearity of the integral that

$$\forall h \in \mathcal{H}: \underline{C}\mathbb{E}_\mathcal{Q}\left[\ell\left(x, h(x)\right)\right] \leq \mathbb{E}_\mathcal{P}\left[\ell\left(x, h(x)\right)\right] \leq \overline{C}\mathbb{E}_\mathcal{Q}\left[\ell\left(x, h(x)\right)\right].$$

To prove necessity, we first assume that $\forall h \in \mathcal{H}: \underline{C}\mathbb{E}_\mathcal{Q}\left[\ell\left(x, h(x)\right)\right] \leq \mathbb{E}_\mathcal{P}\left[\ell\left(x, h(x)\right)\right]$ holds. Let us assume for the moment that $\forall x \in \mathcal{X}: \underline{C}f_\mathcal{Q}(x) > f_\mathcal{P}(x)$. Repeating the argument made to establish sufficiency, the momentary assumption made above implies that $\forall h \in \mathcal{H}: \underline{C}\mathbb{E}_\mathcal{Q}\left[\ell\left(x, h(x)\right)\right] > \mathbb{E}_\mathcal{P}\left[\ell\left(x, h(x)\right)\right]$. Obviously this conclusion contradicts our first assumption. Therefore we have to conclude that $\forall x \in \mathcal{X}: \underline{C}f_\mathcal{Q}(x) \leq f_\mathcal{P}(x)$. Repeating this argument for the second inequality finishes the proof.

To demonstrate that $\underline{C} \leq 1$, we recognize that $\underline{C}f_\mathcal{Q}(x) \leq f_\mathcal{P}(x) \implies \underline{C}\int f_\mathcal{Q}(x) \leq \int f_\mathcal{P}(x)$ which directly establishes this fact. The same argument can be repeated to show that $\overline{C} \geq 1$. □

### A.4 Compositional Data Synthesization Algorithm

---

**Algorithm 1** Compositional data synthesization (CDS) algorithm.

---

```python
frg_to_tpl = defaultdict(list)
tpl_to_frg = defaultdict(list)
env_to_tpl = defaultdict(list)
for seq in dataset:
    #frg, tpl and env are index sets
    for frg in frgs(seq):
        tpl = template(seq, frg)
        env = environment(tpl, window_size)
        #fetch symbols
        frg_s = get_syms(seq, frg, value_type='frg')
        tpl_s = get_syms(seq, tpl, value_type='tpl')
        env_s = get_syms(seq, env, value_type='env')
        #append maps
        frg_to_tpl[frg_s].append(tpl_s)
        tpl_to_frg[tpl_s].append(frg_s)
        env_to_tpl[env_s].append(tpl_s)

frg_list = list(frg_to_tpl)
while True:
    shuffle(frg_list)
    for frg in frg_list:
        tpl_c_list = list(frg_to_tpl[frg])
        shuffle(tpl_c_list)
        for tpl_c in tpl_c_list:
            #get all tpl for frg without tpl_c
            tpl_a_list = [tpl for tpl in tpl_c_list if tpl != tpl_c]
            shuffle(tpl_a_list)
            for tpl_a in tpl_a_list:
                #retrieve templates with same environment as tpl_a
                for tpl_b in env_to_tpl[environment(tpl_a, window_size)]:
                    #retrieve all fragments for tpl_b
                    for frg_b in tpl_to_frg[tpl_b]:
                        if frg_b != frg:
                            ts_tpl_c, ts_frg_c = get_ts_segments(tpl_c, frg)
                            ts_tpl_b, ts_frg_b = get_ts_segments(tpl_b, frg_b)
                            yield insert(ts_tpl_c, ts_frg_b)
```

---

### A.5 Evaluation Metrics

Given $N$ time series in our dataset, with a prediction window of $T$ hours for TSF, the masked mean squared error (MSE) over hourly prediction vectors $\hat{y}_t^n$ is defined as follows:

$$\text{MSE} = \frac{1}{NT} \sum_{n=1}^{N} \sum_{t=1}^{T} ||(y_t^n - \hat{y}_t^n) \odot m_t^n||_2^2 \tag{8}$$

where $m_t^n \in \{0,1\}^{|F|}$ is a mask indicating if the variables in $y_t^n$ were observed or not, and $\odot$ is a component-wise product. In our experiments, $T$ is set to 24 hours.

For each synthetically generated dataset and for the original data, we train three differently seeded models. In our experiments, we report the average MSE of the three training runs ($\overline{\text{MSE}}$) and the corresponding standard deviation (SD). In addition, we report the best performing model ($\text{MSE}^*$) and the 95% confidence interval for the estimation of the evaluation score of $\text{MSE}^*$ on the test set. This is calculated from the sample mean $\overline{\mu}$ and standard deviation $\overline{s}$ of $\text{MSE}^*$ scores on a test set of size $N$ as

$$\text{KI}_{.95} = \overline{\mu} - 1.96 \frac{\overline{s}}{\sqrt{N}}; \overline{\mu} + 1.96 \frac{\overline{s}}{\sqrt{N}}$$

.

### A.6 SOFA Score Definition

The Sepsis-related Organ Failure Assessment (SOFA) is calculated by summing six subscores ranging from 0 to 4. In our setting, we had to recalculate MAP (mean arterial pressure) by SBP and DBP (systolic and diastolic blood pressure), the Horowitz coefficient PaO2/FiO2 by PaO2 and FiO2, but ignored the kind of mechanical ventilation. If no value for calculation in a SOFA subsystem was available, we took a value of 0.

Table 8: SOFA score (Vincent et al., 1996). Abbreviations: CNS = Central nervous system; GCS = Glasgow Coma Scale; MV = mechanically ventilated including CPAP; MAP = mean arterial pressure, UO = Urine output.

| Score | CNS | Cardiovascular | Respiratory | Coagu-lation | Liver | Renal |
|---|---|---|---|---|---|---|
| | GCS | MAP or vasopressors | PaO2/FiO2 (mmHg) | Platelets $(\times 10^3/\mu l)$ | Bilirubin (mg/dl) | Creatinine (mg/dl) or UO |
| +0 | 15 | MAP $\geq$ 70 mmHg | $\geq$ 400 | $\geq$ 150 | $<$ 1.2 | $<$ 1.2 |
| +1 | 13–14 | MAP $<$ 70 mmHg | $<$ 400 | $<$ 150 | 1.2–1.9 | 1.2–1.9 |
| +2 | 10–12 | dopamine $\leq$ 5 $\mu$g/kg/min OR dobutamine (any dose) | $<$ 300 | $<$ 100 | 2.0–5.9 | 2.0–3.4 |
| +3 | 6–9 | dopamine $>$ 5 $\mu$g/kg/min OR epinephrine $\leq$ 0.1 $\mu$g/kg/min OR norepinephrine $\leq$ 0.1 $\mu$g/kg/min | $<$ 200 AND MV | $<$ 50 | 6.0–11.9 | 3.5–4.9 OR $<$ 500 ml/day |
| +4 | $<$ 6 | dopamine $>$ 15 $\mu$g/kg/min OR epinephrine $>$ 0.1 $\mu$g/kg/min OR norepinephrine $>$ 0.1 $\mu$g/kg/min | $<$ 100 AND MV | $<$ 20 | $>$ 12.0 | $>$ 5.0 OR $<$ 200 ml/day |

### A.7 Distributional Properties of Compositional Data in Symbolic Space

Table 9 shows the distributional properties of compositionally synthesized data by computing the pairwise Hellinger distance between symbolic representations of original training data (Tr), original test data (Te), and synthesized data (S). Given two discrete probability distributions $P = (p_1, \ldots, p_k)$ and $Q = (q_1, \ldots, q_k)$, the Hellinger distance is computed by the Euclidean norm of the difference of the square root vectors:

$$H(P, Q) = \frac{1}{\sqrt{2}} \|\sqrt{P} - \sqrt{Q}\|_2. \tag{9}$$

Our experiments show that the distance between original train and original test distributions is an order of magnitude smaller for unigrams than for bigrams or trigrams, and the same relations hold for the distributional distance between synthesized data and original test data. These distributional properties — similar distributions between train and test data on the level of elementary components, here unigrams, and different distributions on the level of compounds, here higher order n-grams — are desired for benchmark data for compositional generalization (Keysers et al., 2020), and are recreated by our compositional data synthesization method.

Table 9: Hellinger distance between distributions of symbolic representations of compositionally synthesized data based on MIMIC-III time series. Symbolization was performed by a random assignment of centroids in input space.

| #Syms | Unigram | | | Bigram | | | Trigram | | |
|---|---|---|---|---|---|---|---|---|---|
| | H(Tr, Te) | H(S, Te) | H(S, Tr) | H(Tr, Te) | H(S, Te) | H(S, Tr) | H(Tr, Te) | H(S, Te) | H(S, Tr) |
| 40 | 0.0908 | 0.0895 | 0.0764 | 0.1612 | 0.1436 | 0.3438 | 0.4007 | 0.2482 | 0.2482 |
| 80 | 0.0237 | 0.0692 | 0.0597 | 0.1488 | 0.1886 | 0.1180 | 0.5231 | 0.5401 | 0.2845 |
| 160 | 0.0348 | 0.0616 | 0.0435 | 0.2712 | 0.2820 | 0.1185 | 0.7271 | 0.7142 | 0.3445 |

### A.8  Qualitative Interpretation of Symbolic Time Series Representations

The goal of this qualitative evaluation is to investigate whether the learned clusters of subsequences of clinical time series can be interpreted as meaningful physiological states. We created dataset versions of MIMIC-III and eICU that are restricted to six features that are measured with high frequency (HR, SBP, DBP, MBP, RR and O2 Saturation). Based on these low-sparsity features, we performed $k$-means clustering on 3-hour blocks of time series subsequences, and assigned symbols to the clusters.

Figure 4 presents the results for $k$-means clustering of neural representations on MIMIC-III data, with $k$ set to 10. Cluster S0 is characterized by elevated heart rate values and can be interpreted as representing the physiological state of tachycardia. Cluster S4 can he interpreted to represent the physiological state of hypertension due to elevated systolic, diastolic, and mean blood pressure. Similar results are obtained by computing clusters based on random centroids in the input domain of MIMIC-III. Figure 5 shows clusters representing tachycardia (S7), hypertension (S0), and tachypnea (S8) due to elevated respiratory rate.

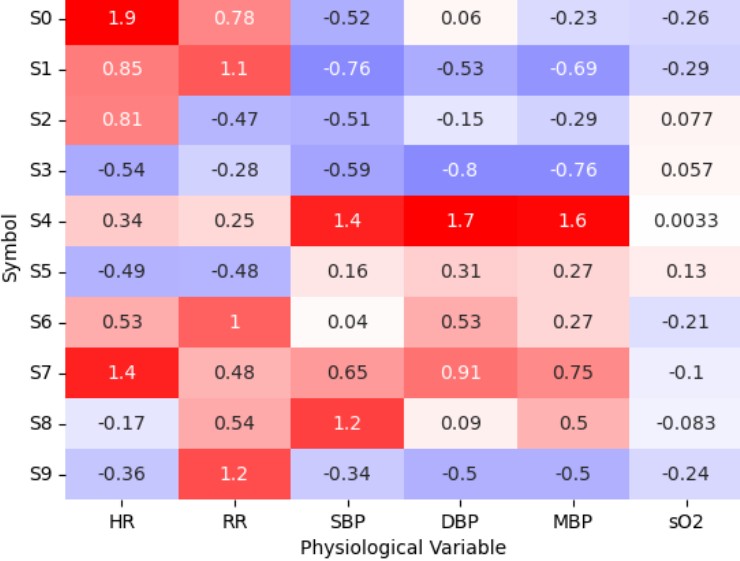

Figure 4: Physiological states learned by time series symbolization based on clustering of neural representations. For example, cluster S0 is characterized by elevated heart rate (HR), representing the physiological state of tachycardia. Cluster S4 can he interpreted to represent the physiological state of hypertension due to elevated systolic, diastolic, and mean blood pressure (SBP, DBP, MBP).

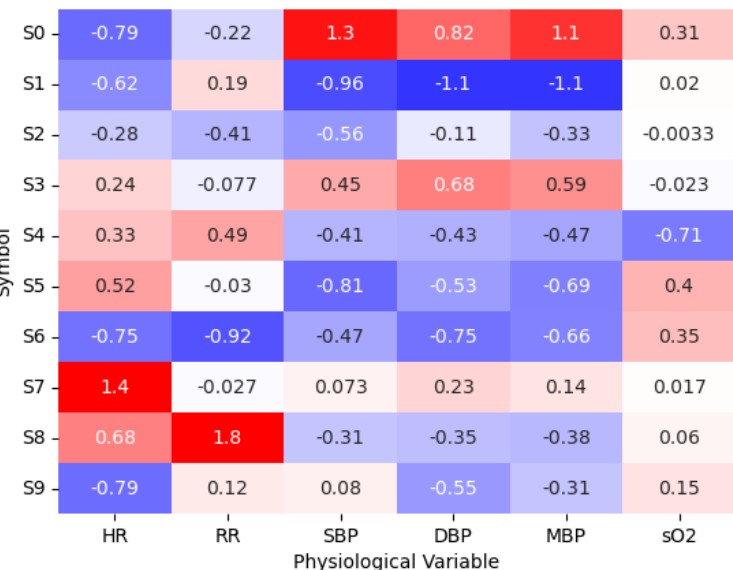

Figure 5: Physiological states learned by time series symbolization based on clustering in input space. Tachycardia is represented by cluster S7, showing elevated HR values. Hypertension is represented by cluster S0, showing elevated BP values. Cluster S8 represents the physiological state of tachypnea due to elevated respiratory rate (RR).

### A.9 Discriminative Score and PCA visualization

For a further quantitative evaluation of our data synthesization algorithm, we take inspiration from work based on generative adversarial networks (GANs) (Yoon et al., 2019; Pei et al., 2021), and calculate a discriminative score as |0.5 - accuracy| for the classification accuracy of distinguishing synthetic examples from original examples. Similar to Yoon et al. (2019), we train an RNN and evaluate the trained classifiers on a held-out testset. The discriminative score is averaged over 10 runs and can be seen in Table 10. Lower discriminative scores are obtained for CDS variants compared to CutMix, indicating higher similarity of original data to compositionally synthesized data than to data synthesized by CutMix.

Table 10: Discriminative scores of CutMix compared to our methods CDS embd and CDS input.

| Model | discriminative score |
|---|---|
| CutMix | 0.384±0.004 |
| CDS embd | 0.301±0.007 |
| CDS input | 0.306±0.008 |

Following Yoon et al. (2019), we furthermore present a qualitative evaluation where we perform PCA on the dense representations of the original data and on each of the generated synthetic data. The visualizations are shown in Figures 6 and 7. We see that CDS covers outlier regions of the original data better than CutMix.

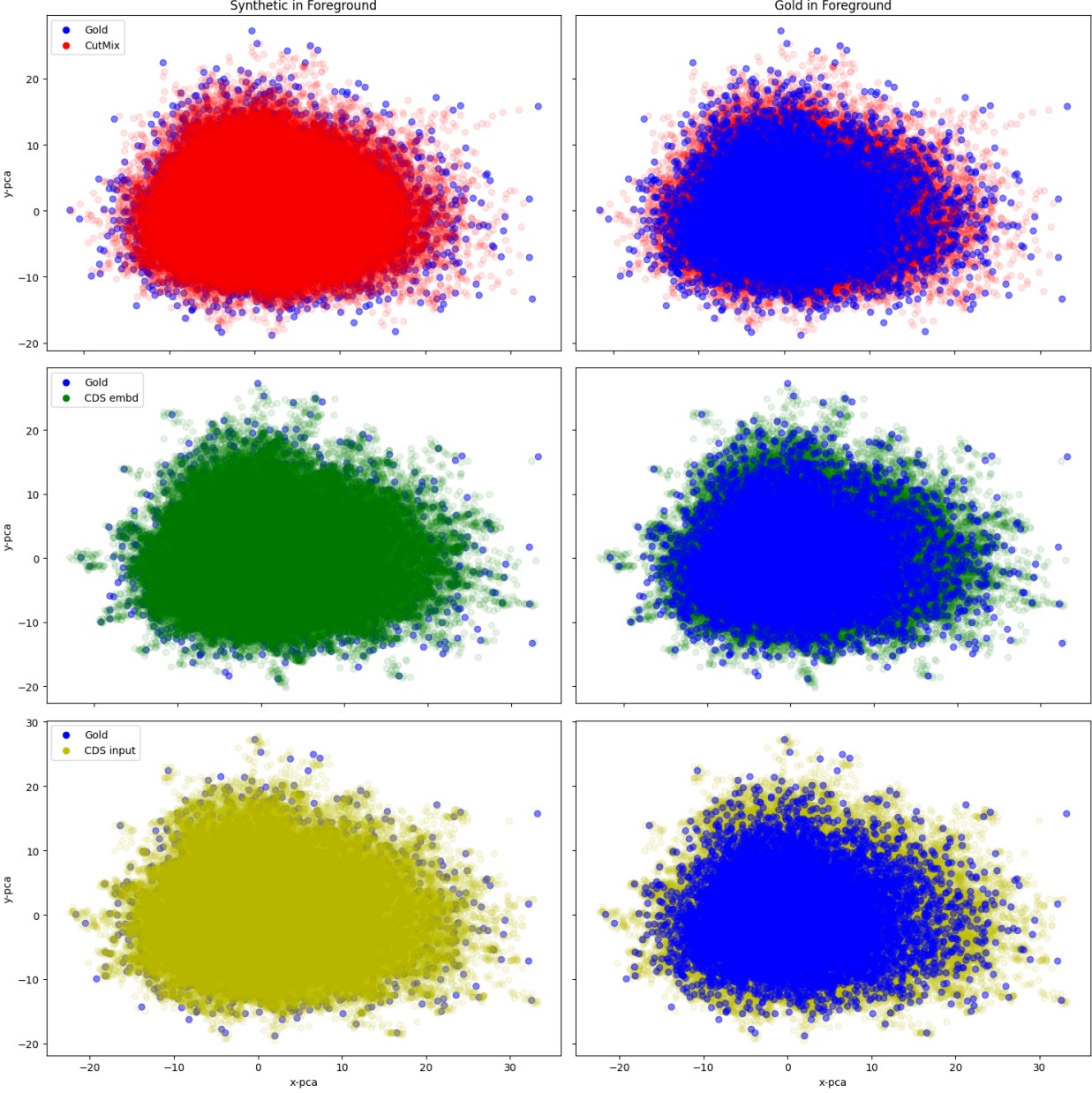

Figure 6: PCA visualisation for MIMIC-III dataset. PCA is applied to the input portion of the dense representation. CutMix covers the main region of the gold data, CDS embd and CDS input however seem to capture the outlier region better.

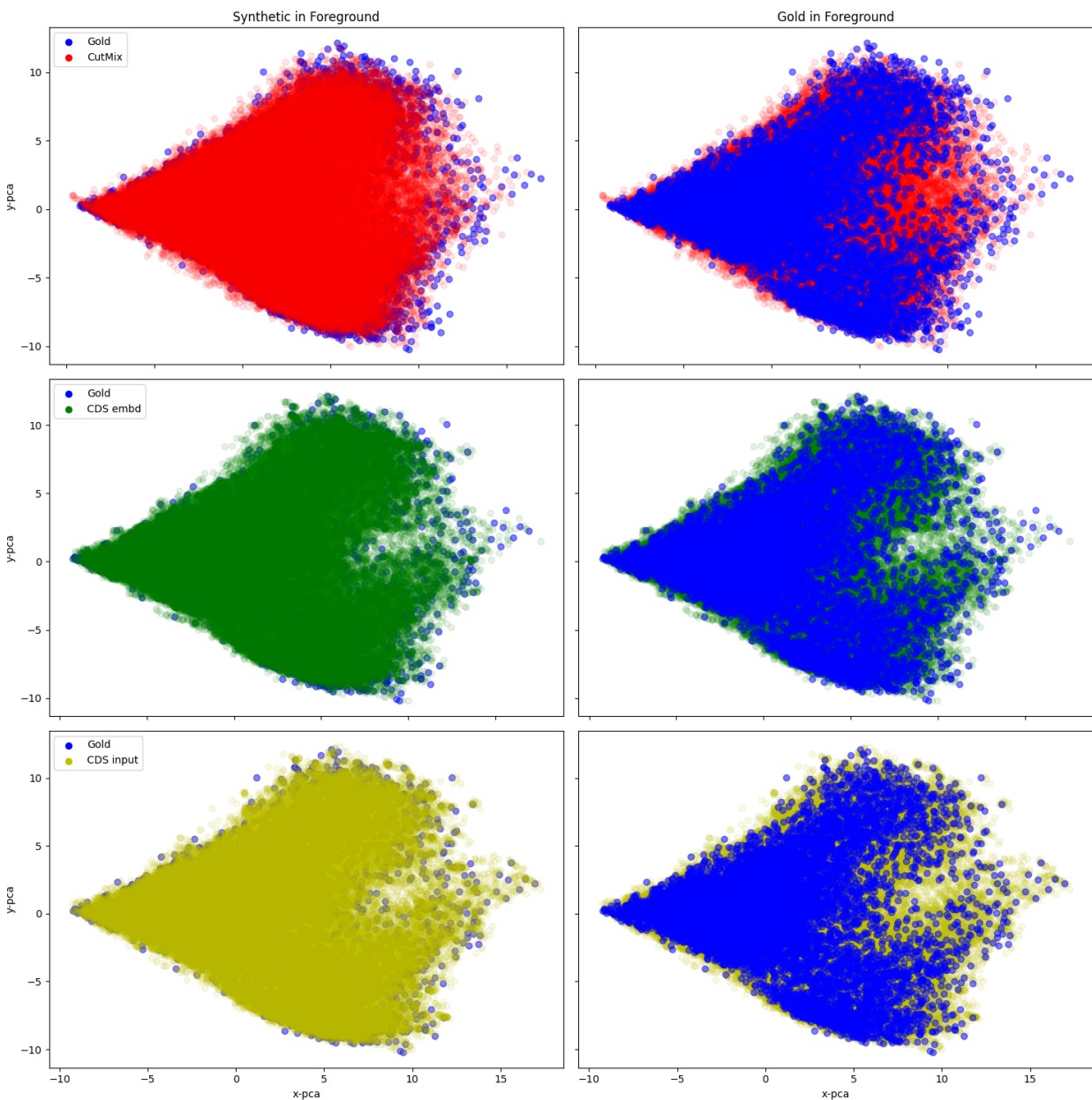

Figure 7: PCA visualisation on masked portions of dense representations for MIMIC-III dataset. Here, CDS embed and CDS input both show the same distinct representation gap on the right side of the scatterplot. This gap is blurred by CutMix.

