# OpenReview forum: "Compositionality in Time Series: A Proof of Concept using Symbolic Dynamics and Compositional Data Augmentation"
_TMLR — Accepted by TMLR_

### Review · Reviewer_hx28 · 2024-11-03

**Summary Of Contributions:**

The authors investigate the utility of synthesized clinical time series data for training time series forecasting (TSF) models using two prominent datasets: MIMIC-III and eICU. They focus on the challenges posed by sparse and irregularly sampled data in intensive care units, from which they extract a comprehensive set of clinical features from patient stays. The authors employ a Transformer-based model with an autoregressive decoder for long-term TSF, evaluating its performance based on various synthesized data versions. A key component of their approach is the compositional data synthesis (CDS) technique, which clusters data in either input or embedding space to capture the natural structure and temporal relationships of clinical measurements before generating synthetic data. This clustering ensures that synthesized data better reflects the inherent patterns of the original dataset. Their experimental results provide insights into the effectiveness of synthesized data for improving model performance, with comparisons between models trained on original and synthetic datasets. The study demonstrates that while non-compositional methods (CutMix) may yield higher error rates, CDS retains useful information from the original data. Overall, the findings highlight the potential of synthetic data in enhancing the robustness of TSF models, ultimately aiming to support better clinical decision-making in healthcare settings.

**Audience:**

Yes

**Broader Impact Concerns:**

Potentially including some statement regarding synthetic data usage. For example, if representation are obtained from MIMIC dataset, there might be some bias towards high density variables or patient metadata. Which might not be aplicable towards other datasets.

**Claims And Evidence:**

Yes

**Requested Changes:**

1) The introduction leans heavily toward describing methods, particularly after the opening paragraph. To strengthen the introduction part, the authors could expand on the importance of compositionality for real-world applications, specifically for time series. A more comprehensive background should address broader contextual points, such as why compositionality is critical in practical scenarios, specifically for time series modeling. Similarly outlining relevant global challenges, existing limitations, and the current literature’s attempts to address these limitations. Highlighting gaps and limitations within the literature would also provide a clearer rationale for the approach taken in this study, setting up a strong foundation for introducing how this work specifically aims to improve upon those areas.  A short introduction of experimental setup could work in the last part of the introduction, but not heavily described.


2) The related work section would benefit from a more direct comparison with prior methods, particularly regarding the obtention of states/clusters which is a significant aspect of this work. For example [1] is a novel alternative approach for states detection, ideally, one could highlight the limitations and advantages of their proposed approach against such work [1] Schäfer P, Leser U. Motiflets: Simple and Accurate Detection of Motifs in Time Series. Proceedings of the VLDB Endowment. 2022 Dec 1;16(4):725-37.  Similarly, the authors tend to over discuss NLP when certainly time series have a large community which is better tailored to their methodology.

3) I would strongly suggest the authors to do not include heavy mathematical notation in the tables caption. Similarly, I would suggest they keep most of the notation within section 3 or methods.

4) The authors did not provide a clear explanation of the methodology or selection criteria for the units of measurement used for each variable. For example, a single laboratory value in the MIMIC dataset may be recorded in various units, which can lead to significant differences in interpretation (e.g., 3.5 mmol/L is not equivalent to 3500 µmol/L). This lack of clarity on unit standardization could impact the reliability and comparability of the results.

5) The authors explain their methodology for handling irregular sampling rates; however, the use of diverse variables with varying frequencies may introduce limitations, such as patient or hospital visit bias, since some features may be recorded more frequently based on specific clinical conditions. These variations could impact the effectiveness of their state detection approach. I recommend that the authors discuss the implications of their methodology regarding sampling rates. Ideally, a brief evaluation stratifying patients or conditions would help quantify these variances, though I do not request extensive experiments due to time constraints.

6) The wording definitions can be improved in terms of unification, for example call ‘states’ or ‘clusters’ across the whole manuscript in a single way when possible. Similarly, ensure that terms like "synthetic data," "compositional data synthesis," and others are used consistently throughout your thesis to avoid confusion.

7) I would strongly suggest to be more explicit with the title. Compositionality in clinical time series is a very broad title. I would suggest the inclusion of synthetic/generation, or time series forecasting.

8) The authors present two clustering methods, the input space clustering and embedding space clustering but do not adequately compare the strengths and weaknesses of each approach. A more detailed discussion on how these methods impact the compositionality of the generated data would be beneficial. Specifically, how does clustering in the input space compare to embedding space in terms of interpretability and capturing underlying physiological states?

9) The authors did not mention the specifics on the thresholds for the definition of healthy variables. At least, a brief discussion of these would be ideal, for example, ALP is higher for children and adolescents due to bone growth. Albumin lower levels are more common in elderly patients. Hemoglobin normal levels are typically higher in males than in females. Creatinine is generally higher in males than in females due to differences in muscle mass.

10) Section 5.1.1 Data contains the storage approach and notation for sparse and irregularly sample time series. From my perspective, this would fit better into the methods section.

11) The authors did not focus on incorporating classification downstream tasks in their analysis of synthesized clinical time series data. Ideally, the authors should have either integrated classification methods into their study or discussed their rationale for omitting this approach.

12) The authors did not test nor properly discuss their synthesis approach for the enhancement of performance on out-of-distribution (OOD), where one could evaluate a model trained on MIMIC, on the eICU dataset. Ideally, the learned compositional representation should be able to generalize towards distinct dataset which are typically the real-world clinical settings.

13) The authors define time series as clinical variables, technically, this is correct, however, by time series, in the medical comunity we would firstly refer to waveforms such as ECG, PPG, EEG. Therefore, it would be great to distinguish between the used variables and waveforms, at least a discussion part into how these would (if would) be integrated into the authors pipeline.


Overall: I would suggest at least discussion of the mentioned topics, and an improved text structure to improve redability (specific text in sections + better handling of math notations). Similarly, additional experiments are a good plus.

**Strengths And Weaknesses:**

Strengths:

1) Innovative creation of synthetic data: The approach utilizes compositinality to synthesize clinical time series data to improve time series forecasting.

2) Comarison of input and embedding states performance: The method properly evaluate two different and interesting states methodologies, e.g. clusters from the input space and clusters from the embedding space.

3) Potential for clinical decision-making: The findings indicate that synthetic data can improve model performance, which may ultimately support better clinical decision-making.


Weaknesses:

See requested changes

---

> ### Author Response · Authors · 2025-01-21
> **Reply to Reviewer hx28**
>
> Thank you for your review! We uploaded a revised version of the manuscript. Please find replies to the numbered requested changes of your review below.
>
> 1. INTRODUCTION: *We extended the introduction to stress the importance of compositionality in our application.*
>
> 2. RELATED WORK: *We extended the related work section with a discussion of pattern recognition in time series analysis.*
>
> 3. NOTATION: *We replaced equations in table captions by textual descriptions.*
>
> 4. UNITS OF MEASUREMENT: *All clinical measurements in our work are standardized Z-scores which makes measurements comparable across different units of measurement.*
>
> 5. IRREGULAR SAMPLING: *Staniek et al. (2024) showed that a mean-value imputation and binning outperform loss-less set-function encodings (Tipirneni et al. (2021), Horn et a. (2020)), thus we follow the former in encoding of clinical time series.*
>
>
> 6. WORDING: *We clarified the connection of latent states to their operationalization as cluster centroids in the introduction.*
>
> 7. TITLE: *We changed the title into "Compositionality in Time Series: A Proof of Concept using Symbolic Dynamics and Compositional Data Augmentation"*
>
> 8. CLUSTERING: *In ablation experiments, we found similar performance for symbolization by clustering in input domain versus clustering in the space of embeddings.*
>
> 9. THRESHOLDS: *The newly added downstream task of prediction of SOFA scores (see below) is based on thresholding clinical measurements for six organ systems.*
>
> 10. DATA: *We consolidated these subsections into one section called "Data, Preprocessing, and Models".*
>
> 11. DOWNSTREAM TASK: *Thank you very much for this recommendation. We added an experimental evaluation of randomized  (CutMix) and compositional data augmentation (CDS) on the downstream task of prediction of SOFA scores from thresholds on clinical measurements (see Vincent et al. (1996)). The experimental results show wins of CDS over CutMix and over training from original data. A description and analysis of this experiment is given in the updated paper. he results are repeated below for convenience, showing significant decreases in MSE (smaller is better) for models
> trained on compositionally synthesized data (CDS) that are 5, 10, or 20 times larger than the original
> training set. Prediction for models trained on randomized-based
> data synthesization (CutMix) does not significantly improve over models trained on original data, despite the size of
> the synthetic dataset being 20 times larger.*
>
> | Training data    | Dataset size | MIMIC-III    | eICU |
> | :---        |    :---  |     :--- |   :--- |
> | original      |   1x  |     4.168 |   2.518 |
> | CDS     |   1x  |     4.066 |   2.536 |
> | CDS     |   5x  |     3.560 |   2.397 |
> | CDS     |   10x  |     3.332 |   2.347 |
> | CDS     |   20x  |     3.194 |   2.322 |
> | CutMix    |   20x  |     4.063 |   2.583 |
>
> 12. OOD: *Compositional generalization is motivated in our work in domain adaptation theory, not OOD generalization. OOD assumes a general shift from the train distribution to an unknown test distribution where P_train(X,Y) != P_test(X,Y), while DA assumes some prior knowledge about the test distribution and milder conditions like covariate shift where (P_train(Y|X) = P_test(Y|X) and P_train(X) != P_test(X)). The latter conditions are satisfied in our set of partitioning on data set into train, dev, and test portions.*
>
>
> 13. WAVEFORMS: *We clarified in the related work section that we focus on time series of discrete measurement points, not continuous waveforms.*

---

### Review · Reviewer_Xak5 · 2024-11-12

**Summary Of Contributions:**

The paper presents a framework for assessing compositionality in clinical time series by decomposing data into symbolic representations of latent physiological states. The authors propose two tests, based on domain adaptation theory, to evaluate whether clinical time series data can be accurately represented and generated through compositional principles. The effectiveness of this synthetic data is evaluated by comparing model performance on compositional versus random synthetic data. Results indicate that models trained on compositional synthetic data perform comparably to those trained on the original data, suggesting that the synthetic data retains relevant structural patterns. The authors suggest that this approach may be generalisable to other types of time series data.

**Audience:**

Yes

**Broader Impact Concerns:**

This is a medical/clinical paper yet is very fundamental. There is nothing that I see that would warrant a specific impact statement.

**Claims And Evidence:**

Yes

**Requested Changes:**

Apart from fixing the above minor things, here are the changes I recommend:

- Critical: The authors should discuss the impact of using fixed-length (three-hour) segments, including how this choice might limit the ability to capture variable-duration clinical events. Highlighting trade-offs in this segmentation approach would provide a clearer understanding of the compositional assumptions.
- Strengthening: Additional experiments/tasks such as classification (e.g., detecting sepsis or other available label) alongside time series forecasting would provide more comprehensive evidence for the versatility of the synthetic data in clinical applications.
- Strengthening: Including comparisons with baseline methods like traditional data augmentations (e.g., random time shifts or scaling) would provide stronger evidence for the uniqueness and benefits of the compositional approach.
- Strengthening: A discussion on the interpretability and potential clinical significance of the symbolic states generated through clustering would improve the paper.
- Strengthening: Conducting a preliminary analysis to see if certain symbolic clusters correlate with known clinical markers or conditions

I've listed the additional experiments as *strengthening* here, yet want to reiterate that the experiments are marginal and the paper would significantly benefit from more experiments. It is finely balanced between that and the fact the paper is very interesting and well-written.

**Strengths And Weaknesses:**

## Strengths
- Proposes a novel framework for testing compositionality in clinical time series data
- Introduces symbolic representation through clustering to enable compositional data synthesis
- Develops two domain adaptation-based tests to assess compositionality empirically
- Uses compositional synthetic data to train time series forecasting models, showing comparable performance to models trained on original data
- Suggests generalisability of the approach to other time series data types

## Weaknesses
- The framework relies on fixed-length (three-hour) segments, which may overlook variable-duration events that are common in clinical data. This segmentation approach risks losing important physiological nuances that variable-length segments might capture.
- While symbolic states are generated through clustering the clinical relevance of these states is not validated. Without expert interpretation, it is unclear whether these symbols represent meaningful physiological conditions or are artifacts of the clustering process.
- The synthetic data and compositional tests are not evaluated in the context of a specific clinical application. This limits the practical relevance of the findings, as the symbolic representations and synthetic sequences are not tested for direct utility in clinical decision-making or prediction. The authors acknowledge this, yet it remains something that would add significant weight to the paper and should be feasible given the current experiments and datasets used.
- The experiments are fairly basic. There are a number of quick add-on experiments that can be done – the authors could include a more diverse set of tasks, such as classification (e.g., distinguishing between patient states or detecting specific conditions like sepsis), or including additional baseline methods, such as data generated through more traditional augmentation techniques (e.g., random time shifts or scaling of real data). That would provide a stronger context for assessing the uniqueness of the compositional approach

## Minor things
- Only the first equation numbered, not subsequent ones
- Figure 1 caption: “hearth and lung” -> “heart and lung”
- Abstract: “framework work” -> “framework”
- “Transformer” is inconsistently capitalised, see in particular section 5.1.2 which uses both. Recommend sticking to one or the other throughout
- dobutamin -> dobutamine
- In the Szabó reference: Stanford Encylopedia of Philosophy -> Stanford Encyclopedia of Philosophy
- Figure 3: “with with” -> “with”

---

> ### Author Response · Authors · 2025-01-21
> **Reply to Reviewer Xak5**
>
> Thank you for your review! We uploaded a revised version of the manuscript. Please find replies to your requested changes below.
>
> REQUESTED CHANGE: Critical: The authors should discuss the impact of using fixed-length (three-hour) segments, including how this choice might limit the ability to capture variable-duration clinical events. Highlighting trade-offs in this segmentation approach would provide a clearer understanding of the compositional assumptions.
>
> *REPLY: Our compositional data synthesization algorithm (CDS, Section 4.2 and Appendix A.4) induces variable-duration time series segments as a by-product of the data augmentation procedure. The so-called "fragments", i.e., sequences of symbols that can be exchanged into new contexts, can consist of multi-symbol sequences, corresponding to a concatenation of multiple 3-hour time series segments. The choice of the minimal length of 3 hours is a compromise between very sparse 1-hour windows where over 90% of the measurements need to be imputated, and segments with a better descriptive power of temporal patterns.*
>
> REQUESTED CHANGE: Strengthening: Additional experiments/tasks such as classification (e.g., detecting sepsis or other available label) alongside time series forecasting would provide more comprehensive evidence for the versatility of the synthetic data in clinical applications.
>
> *REPLY: Thank you very much for this recommendation. We added an experimental evaluation of randomized  (CutMix) and compositional data augmentation (CDS) on the downstream task of prediction of SOFA scores from thresholds on clinical measurements (see Vincent et al. (1996)). The experimental results show significant wins of CDS over CutMix and over training from original data. A description and analysis of this experiment is given in the updated paper. The results are repeated below for convenience, showing significant decreases in MSE (smaller is better) for models
> trained on compositionally synthesized data (CDS) that are 5, 10, or 20 times larger than the original
> training set. Prediction for models trained on randomized-based
> data synthesization (CutMix) does not significantly improve over models trained on original data, despite the size of
> the synthetic dataset being 20 times larger.*
>
> | Training data    | Dataset size | MIMIC-III    | eICU |
> | :---        |    :---  |     :--- |   :--- |
> | original      |   1x  |     4.168 |   2.518 |
> | CDS     |   1x  |     4.066 |   2.536 |
> | CDS     |   5x  |     3.560 |   2.397 |
> | CDS     |   10x  |     3.332 |   2.347 |
> | CDS     |   20x  |     3.194 |   2.322 |
> | CutMix    |   20x  |     4.063 |   2.583 |
>
> REQUESTED CHANGE: Strengthening: Including comparisons with baseline methods like traditional data augmentations (e.g., random time shifts or scaling) would provide stronger evidence for the uniqueness and benefits of the compositional approach.
>
> *REPLY: According to Guo et al. (2023) and Yang & Desell (2022), traditional transformation-based augmentations for time series like flipping, scaling, or methods based on Fourier transforms, require human understandings of the data and show unstable performance. Instead, they recommend mix-based data augmentation techniques like CutMix for multivariate time series and show very strong results. CutMix is thus a strong baseline and it fits our work ideally since it can be seen as a randomized variant of our compositional data augmentation method.*
>
> REQUESTED CHANGE: Strengthening: A discussion on the interpretability and potential clinical significance of the symbolic states generated through clustering would improve the paper.
>
> *REPLY: A qualitative interpretation of symbolic time series representations has been given in Appendix A.7 in the original submission, now Appendix A.8 in the revised manuscript. These interpretations are based on dataset versions of MIMIC-III and eICU that are restricted to six features that are measured with high frequency (HR, SBP, DBP, MBP, RR and O2 Saturation). We find clusters corresponding to tachycardia, hypertension, or tachypnea.*
>
>
> REQUESTED CHANGE: Strengthening: Conducting a preliminary analysis to see if certain symbolic clusters correlate with known clinical markers or conditions.
>
> *REPLY: Unfortunately, the clinical variables relevant for early prediction of SOFA/Sepsis are sparsely measured lab values that do not build a good data basis for clustering algorithms. We thus restricted a qualitative interpretation of symbolic custers to a small number of densely measured vital signals (see reply above).*

---

> > ### Comment · Reviewer_Xak5 · 2025-02-03
> >
> > I thank the authors for their responses.
> >
> > First, they clarified my misconception about the fixed-length segments and have added more experiments with SOFA using Cutmix and CDS.
> >
> > While I understand that traditional baselines may perform poorly and be unstable, including them would still provide valuable context for comparison. I do note that Cutmix provides a solid baseline nonetheless.
> >
> > The authors provided a clear explanation for why additional clustering analysis isn't feasible due to the sparsity of lab measurements.
> >
> > However, the typographical and formatting errors I noted (under "minor things") remain present in the revised manuscript and have not been addressed.
> >
> > In conclusion, the authors have made moderate improvements to address the major requested changes and some concerns are fully addressed. The minor corrections remain unaddressed - while this doesn't affect the scientific rigour of the paper, it does impact its overall quality. I do share reviewer PVFn's view that TimeGAN comparisons would significantly strengthen the paper and urge the authors to add them, but I do not see that as an absolute hindrance to acceptance.

---

> > > ### Author Response · Authors · 2025-02-03
> > > **Reply to Reviewer Xak5**
> > >
> > > Thanks again for your reply. Please find our answers below.
> > >
> > > QUESTION: The typographical and formatting errors I noted (under "minor things") remain present in the revised manuscript and have not been addressed.
> > >
> > > *ANSWER: We are terribly sorry about this omission. In the latest revision of the paper, all "minor things" are fixed.*
> > >
> > > QUESTION: I do share reviewer PVFn's view that TimeGAN comparisons would significantly strengthen the paper and urge the authors to add them, but I do not see that as an absolute hindrance to acceptance.
> > >
> > > *ANSWER: We would like to stress that the goal of our work was not to present an optimal data augmentation method for time series, but to analyze time series from the perspective of compositional data generation, and to use empirically testable criteria on compositional data synthesization as a proof-of-concept for the validity of this perspective. The fact that GAN-generated data synthesized achieve better empirical results than compositionally synthesized data does not provide any insights into the compositional nature of time series. This can only be achieved in a systematic ablation experiment where the contribution of compositionality in data augmentation is systematically removed and replaced by random sampling - this is exactly what CutMix provides. A comparison to GAN-based baselines makes sense in works that are based on GANs, however, consequently these works do not compare against CutMix as baselines for time series synthesization.*

---

### Review · Reviewer_PVFn · 2025-01-20

**Summary Of Contributions:**

- The authors propose an interesting approach for synthetic time-series data generation via compositionality.
- The proposed method is identifying symbolic representation sequences from the original time-series data to generate synthetic time-series data (based on the learned symbolic representation distributions).
- The experimental results show that the proposed method performs better than the simple baseline.

**Audience:**

Yes

**Claims And Evidence:**

No

**Requested Changes:**

**1. Experiments**
- What is "x" here? How is this "x" related to "S"? How do you encode input to make "x"?
- Do you have some "qualitative results"? Like showing the synthetically generated samples and compared with the original samples?

**2. Baseline**
- As we know, there are various time-series generation methods (including TimeGAN and the papers which cite this paper).
- However, the authors only compare with CutMix which is actually not developed for time-series synthetic data generation models.
- In order to provide the advantages of the proposed method, the authors should utilize more advanced time-series synthetic data generation methods than the current baselines.

**3. Metrics**
- For synthetic data generation, there are various other metrics including "discriminator approach" (train another classifier that tries to distinguish between generated data and original data.
- Also, we can do some feature analysis as well.
- Currently, I think the ideas are interesting and promising but the (empirical) supports are not enough.

**Strengths And Weaknesses:**

Strength:
- The proposed method is interesting and intuitive.
- The paper is well written.

Weakness:
- The experimental results are less convincing.
- The metrics are not enough.
- The baselines are not enough.
- Need more qualitative results.

---

> ### Author Response · Authors · 2025-01-21
> **Reply to Reviewer PVFn**
>
> Thank you for your review! We uploaded a revised version of the manuscript. Please find replies to your requested changes below.
>
> REQUESTED CHANGE: Experiments: What is "x" here? How is this "x" related to "S"? How do you encode input to make "x"?
>
> *REPLY: S is the "raw" input time series which is encoded into a densified format $x$ where every timestep is a vector of feature values representing one hour. We construct this vector by choosing the first observed value during the represented hour for each feature. If no value was observed, we impute zero which corresponds to the mean value due to standardization of the data. This was described in the original submission and is made more explicit in the revised manuscript.*
>
> REQUESTED CHANGE: Experiments: Do you have some "qualitative results"?
>
> *REPLY: A qualitative interpretation of symbolic time series representations can be found in Appendix A.7 in the original submission, now Appendix A.8 in the revised manuscript. These interpretations are based on dataset versions of MIMIC-III and eICU that are restricted to six features that are measured with high frequency (HR, SBP, DBP, MBP, RR and O2 Saturation). We find clusters corresponding to tachycardia, hypertension, or tachypnea. Furthermore, we discuss the distributional properties of compositionally generated data in Appendix A.7 of the revised paper.*
>
> REQUESTED CHANGE: Baseline: The authors only compare with CutMix which is actually not developed for time-series synthetic data generation models.
>
> *REPLY: According to Guo et al. (2023) and Yang & Desell (2022), traditional transformation-based augmentations for time series like flipping, scaling, or methods based on Fourier transforms, require human understandings of the data and show unstable performance. Instead, they recommend mix-based data augmentation techniques like CutMix for multivariate time series and show very strong results. CutMix is thus a strong baseline and it fits our work ideally since it can be seen as a randomized variant of our compositional data augmentation method.*
>
> REQUESTED CHANGE: Metrics: For synthetic data generation, there are various other metrics including "discriminator approach" (train another classifier that tries to distinguish between generated data and original data.
>
> *REPLY: Thank you for this pointer. The TSTR (Train on Synthetic, Test on Real) and TRTS (Train on Real, Test on Synthetic) evaluations in Esteban et al. (2017) are related to Test 1 and Test 2 in our work, respectively. However, our tests are motivated in domain adaptation theory and allow to draw general conclusions about the distributional similariy of synthetic and original data. We included the respective citations in the revised manuscript.*
>
> REQUESTED CHANGE: Metrics: Currently, I think the ideas are interesting and promising but the (empirical) supports are not enough.
>
> *REPLY: We added an experimental evaluation of randomized  (CutMix) and compositional data augmentation (CDS) on the downstream task of prediction of SOFA scores from thresholds on clinical measurements (see Vincent et al. (1996)). The experimental results show significant wins of CDS over CutMix and over training from original data. A description and analysis of this experiment is given in the updated paper. The results are repeated below for convenience, showing significant decreases in MSE (smaller is better) for models
> trained on compositionally synthesized data (CDS) that are 5, 10, or 20 times larger than the original
> training set. Prediction for models trained on randomized-based
> data synthesization (CutMix) does not significantly improve over models trained on original data, despite the size of
> the synthetic dataset being 20 times larger.*
>
> | Training data    | Dataset size | MIMIC-III    | eICU |
> | :---        |    :---  |     :--- |   :--- |
> | original      |   1x  |     4.168 |   2.518 |
> | CDS     |   1x  |     4.066 |   2.536 |
> | CDS     |   5x  |     3.560 |   2.397 |
> | CDS     |   10x  |     3.332 |   2.347 |
> | CDS     |   20x  |     3.194 |   2.322 |
> | CutMix    |   20x  |     4.063 |   2.583 |

---

> > ### Comment · Reviewer_PVFn · 2025-01-30
> > **Response to the rebuttal**
> >
> > Thanks a lot for your rebuttal.
> > I carefully read the entire rebuttal as well as the revised manuscript.
> >
> > However, actually, most of my main concerns are not well addressed.
> >
> > 1. Qualitative results
> > - Here, the qualitative results that I requested is more like examples of "generated synthetic time-series data (raw data in dense format)" because I would like to check how the generated samples are similar with real time-series data.
> > - Appendix A.8 is "aggregated qualitative analyses". If the authors can provide "raw synthetic samples in a dense format", that would be good for me to further evaluate the paper.
> >
> > 2. Baselines
> > - Unfortunately, the authors did not compare with TimeGAN and its variants which have the same objective (synthetic time-series data generation).
> > - I think just cutmix and simple augmentation methods are not enough baselines because there are many more algorithms and methods for synthetic time-series data generation. (Please check TimeGAN and other works that cite TimeGAN).
> >
> > 3. Metrics
> > - Unfortunately, the authors did not increase the metrics of the synthetic data evaluation.
> > - Note that in generative model field, there is no "one" standard evaluation metric unlike predictive model fields.
> > - Therefore, to verify whether the generated synthetic data is valuable, we need to use various evaluation metrics to verify.
> > - I suggested some metrics in the previous reviews.
> >
> > In general, I still did not convince on the empirical results due to the above reasons.
> > Thus, I will maintain my original evaluation on this paper.
> >
> > Thanks!

---

> > > ### Author Response · Authors · 2025-02-03
> > > **Reply to Reviewer PVFn**
> > >
> > > Thanks again for your reply. Please find our answers below.
> > >
> > > QUESTION: 1. Qualitative results: The qualitative results that I requested is more like examples of "generated synthetic time-series data (raw data in dense format)" because I would like to check how the generated samples are similar with real time-series data.
> > >
> > > *ANSWER: We cannot share MIMIC-III and eICU data which are restricted under the PhysioNet Credentialed Health Data Use Agreement, unless in the form of aggregated analyses. However, we can wrap up our code in order to allow researchers with credentialed access to extract raw and synthetic time series data from these resources. This can be done quickly upon request.*
> > >
> > > QUESTION: 2. Baselines: Unfortunately, the authors did not compare with TimeGAN and its variants which have the same objective (synthetic time-series data generation).
> > >
> > > *ANSWER: We would like to stress that the goal of our work was not to present an optimal data augmentation method for time series, but to analyze time series from the perspective of compositional data generation, and to use empirically testable criteria on compositional data synthesization as a proof-of-concept for the validity of this perspective. The fact that GAN-generated data synthesized achieve better empirical results (or not) than compositionally synthesized data does not provide any insights into the compositional nature of time series. Such an insight can only be achieved by a systematic ablation where the contribution of compositionality in data augmentation is systematically removed and replaced by random sampling - this is exactly what CutMix provides. A comparison to GAN-based baselines makes sense in works that are based on GANs, however, consequently these works do not compare against CutMix as baselines for time series synthesization.*
> > >
> > > QUESTION: 2. Metrics: Unfortunately, the authors did not increase the metrics of the synthetic data evaluation. Note that in generative model field, there is no "one" standard evaluation metric unlike predictive model fields. Therefore, to verify whether the generated synthetic data is valuable, we need to use various evaluation metrics to verify. I suggested some metrics in the previous reviews.
> > >
> > > *ANSWER: Our work is evaluated according to three quantitative metrics - train on synthetic, test on real (Section 5.2.1), train on real, test on synthetic (Section 5.2.2.), downstream task of SOFA score prediction (Section 5.2.3) - and two qualitative metrics - distributional properties (Appendics A.7), clinical interpretation of symbolic representations (A.8). However, we will additionally compute a quantitative evaluation according to discriminative scores, and a qualitative evaluation according to a PCA visualization as done in the TimeGAN paper. These evaluations will be available in the next days.*

---

> > > > ### Author Response · Authors · 2025-02-05
> > > > **Reply to Reviewer PVFn, updated**
> > > >
> > > > Dear Reviewer PVFn,
> > > >
> > > > we uploaded a revised version of the manuscript that contains a quantitative evaluation with respect to the discriminative score, and a qualitative evaluation according to a PCA visualization, inspired by works on GAN-based data synthesization (Yoon et al., 2019; Pei et al., 2021), in Appendix 9.

---

> > > > > ### Comment · Reviewer_PVFn · 2025-02-07
> > > > > **Response to the rebuttal**
> > > > >
> > > > > Thanks a lot for your answers and additional results in the revised manuscript.
> > > > > Those additional results are beneficial for this paper.
> > > > > I will finalize my evaluation based on this latest revised manuscript.

---

### Comment · Action_Editor_NvEr · 2024-12-11
**Apologies for the delay**

Hello authors,

I wanted to extend an apology for the delay on our end. It has been difficult getting a third reviewer to spend an adequate amount of time to review your work. I have just selected a new reviewer who I hope will be able to respond and complete their review in a timely manner. Thank you for your patience!

Best,
AE

---

> ### Author Response · Authors · 2024-12-20
> **Re: Delay in reviewing and rebuttal**
>
> Dear action editors and reviewers,
> We wanted to inform you that we will publish our rebuttal as soon as the third review has been submitted. However, due to the delay in the third review and due to possible delays on our side during the upcoming holidays, we would like to ask you to wait until we have been able to publish our rebuttal before making your final decision.
>
> Yours sincerely,
> the team of authors

---

> > ### Comment · Action_Editor_NvEr · 2024-12-20
> > **This is appropriate; Deepest apologies on our end**
> >
> > I believe this to be a reasonable request. Again, I am very sorry about the delays we've had securing a third review for you. We've had a lot of difficulty recruiting someone to fully accept the responsibility.
> >
> > I will request from the acting editors in chief to extend the decision period until after the holiday season.
> >
> > Best,
> > AE

---

> > > ### Author Response · Authors · 2025-01-09
> > > **Rebuttal and discussion with reviewers Xak5 and hx28**
> > >
> > > Dear action editor and reviewers Xak5 and hx28,
> > >
> > > we would like to inform you that we addressed all questions raised by the first two reviewers and would be ready to post a revised version of the paper and a comment addressing the questions. Please advise if we should still wait for the delayed third review or if we should start the discussion now.
> > >
> > > Best regards,
> > > the team of authors

---

> > > > ### Comment · Action_Editor_NvEr · 2025-01-14
> > > > **Unfortunately, we'll need to wait**
> > > >
> > > > Hi,
> > > >
> > > > I'm again so sorry that we have to wait for a third reviewer. It's not ideal but is the recommendation from the acting editors in chief. I am re-doubling my efforts this week to personally contact the reviewers I've invited to see if we can't get this discussion started.
> > > >
> > > > Best,
> > > > Taylor

---

### Decision · Action_Editor_NvEr · 2025-02-26

**Recommendation:** Accept as is

**Comment:**

The authors have updated the paper sufficiently in response to the reviewers concerns and suggestions. I feel that it is ready as is for publication. I will caution the authors however against some of the broader concerns raised in the reviews about adequate positioning of the contributions of this work. There are some considerations of more thoroughly discussing prior work before submitting the camera ready version of the work. Additionally, I recommend that the authors follow through with their promise of making their code available, as they have promised.

**Audience:**

All reviewers agreed that this work will find audience among the ML Research, and more importantly the TMLR, community. As stated above, there may be reservations from some about fully implementing CDS and the compositional data generation approach since the paper does not exhaustively compare against all prior approaches. Again, this is not a reason to hold the paper from the community as established in TMLR guidelines.

**Claims And Evidence:**

This paper presents a new approach for synthetic time-series data generation via compositionality, identifying symbolic representation subsequences in the native time series using them to generate new instances of the time series. The paper also introduces two tests to evaluate the time series and their suitability for seeding the synthetic generation. It is shown that the proposed test retains useful information from the original data where prior approaches admit higher error.

There were some concerns that the paper does not adequately test against GAN based approaches. However, the claims of the paper do not attempt to establish superiority over these methods. The primary evaluation is between the proposed CDS and the prior effort CutMix. While this does not address the primary concerns of rigorous evidence in comparison with GANs, it is beyond the proposed scope of the work. This does not disqualify the work from satisfying the TMLR criteria, however it must be stated that it does lessen the potential impact of the work as has been put forward by the authors.